# Harnessing Heterogeneous Statistical Strength for Personalized Federated Learning via Hierarchical Bayesian Inference

**Mahendra Singh Thapa** [1]   **Rui Li** [1]

## Abstract

Personalized federated learning (PFL) based on Bayesian approach tackle the challenges from statistical heterogeneity of client data by computing a personalized posterior distribution over the parameters of each client's local model and constructing a global distribution by aggregating the parameters of these personalized posteriors. However, the heuristic aggregation methods introduce strong biases and result in global models with poor generalization. We thus propose a novel hierarchical Bayesian inference framework for PFL by specifying a conjugate hyper-prior over the parameters of the personalized posteriors. This allows us to jointly compute a global posterior distribution for aggregation and the personalized ones at local level. This hierarchical Bayesian inference framework achieves elegant balance between local personalization and global model robustness. Extensive empirical study shows that by effectively sharing the heterogeneous statistical strength across the local models while retaining their distinctive characteristics, our framework yields state-of-the-art performance. We also show that existing Bayesian PFLs are special cases of our framework.

## 1. Introduction

Federated learning (FL) emerges as a promising framework that enables collaborative model training across decentralized client devices while preserving data privacy (McMahan et al., 2017). This approach demonstrates significant potential in various real-world applications, such as healthcare (Yang et al., 2021; Dayan et al., 2021; Yang et al., 2024), recommendation systems (Muhammad et al., 2020; Yuan

[1]College of Computing and Information Sciences, Rochester Institute of Technology, Rochester, New York, USA. Correspondence to: Rui Li <rxlics@rit.edu>.

*Proceedings of the 42$^{nd}$ International Conference on Machine Learning*, Vancouver, Canada. PMLR 267, 2025. Copyright 2025 by the author(s).

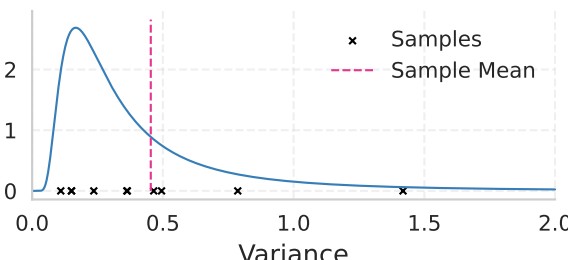

*Figure 1.* In Bayesian PFL, variance parameters of personalized distributions often follow skewed distributions. In such cases, the MLE/MAP estimate (e.g., arithmetic mean) is obviously a poor estimate.

et al., 2024), and mobile applications (Hard et al., 2018; Wu et al., 2020; Geng et al., 2024). However, FL still faces substantial challenges, mainly due to statistical heterogeneity and limited data availability on individual clients (Ye et al., 2023).

The Bayesian-based personalized FL (PFL) approaches focusing on learning personalized local models for individual clients perform Bayesian inference to derive personalized posterior distributions of local model parameters (Zhang et al., 2022; Liu et al., 2023; Bhatt et al., 2024; Makhija et al., 2024). The server then aggregates the parameters of these personalized posteriors to construct a global distribution. Specifically, the construction of the global distributions in these approaches typically involves arithmetic aggregation of personalized posterior parameters as the global distribution parameters (Zhang et al., 2022) or approximation of the global distributions as the product of the personalized ones (Liu et al., 2023), as shown in Figure 2. The global distributions are then transmitted back to all clients as a local regularization for the next training iteration. These simple yet rigid aggregation schemes at the global level introduce strong constraints that obscure unique characteristics distinguishing the heterogeneous local data and lead to biased estimates of the global models with poor generalization, as illustrated in Figure 1.

There are Bayesian-based FL approaches focusing on constructing robust global models, such as computing global dis-

tributions by fitting standard distributions to clients' point-estimated local models (Thorgeirsson & Gauterin, 2020; Chen & Chao, 2021), or constructing point-estimated global models from local estimates (Yurochkin et al., 2019) through nonparametric Bayesian methods. Since these approaches only obtain generic global models without personalized distributions for each client, they are less effective at capturing the diversity of non-IID data in real-world FL settings.

We thus propose a novel hierarchical Bayesian inference framework for PFL, addressing these challenges by balancing between local personalization and global model robustness, as demonstrated in Figure 2. Specifically, we define a conjugate hyper-prior distribution over the parameters of the personalized posterior distributions, allowing us to perform joint Bayesian inference at both the global level and the local level. By computing a higher level posterior distribution over the personalized posterior parameters on the server, instead of summarizing them through arithmetic means, our framework achieves an elegant compromise. Estimates based on the global-level posterior dependencies between the personalized distribution parameters are "shrunk" together, so that the local models share the heterogeneous statistical strength of each other, while retaining their distinctive characteristics. Our theoretical analysis shows that our framework recovers existing Bayesian PFL objectives when additional constraints are introduced, which subsumes them as special cases. Our extensive empirical analysis shows that the proposed hierarchical Bayesian inference PFL results in robust global and local models with superior performance.

## 2. Related Works

### 2.1. FL and Bayesian FL

In the seminal work of FL, FedAvg (McMahan et al., 2017) trains a local model on clients, aggregates them on the server considering only model parameters to construct a global model, and broadcasts this global model to all clients for subsequent training. Subsequent research efforts have focused on improving convergence (Karimireddy et al., 2020), ensuring privacy guarantees (Agarwal et al., 2018), enhancing communication efficiency (Reisizadeh et al., 2020), and refining client aggregation (Zhu et al., 2021). Furthermore, several Bayesian-based approaches have been explored to obtain a global model by fitting Gaussian or Dirichlet distributions to clients' point-estimated local models to define posterior distributions over global model parameters (Chen & Chao, 2021; Thorgeirsson & Gauterin, 2020). An alternative approach focuses on defining the global posterior as the product of local posteriors under a uniform prior (Al-Shedivat et al., 2021). Additionally, researchers have investigated nonparametric Bayesian methods to construct point-estimated global models from local estimates

(Yurochkin et al., 2019). However, these methods are less effective in capturing the inherent statistical heterogeneity present across clients.

### 2.2. Personalized FL

Personalized FL aims to introduce a personalized model for individual clients to tackle the heterogeneity that exists in the FL environment. Several perspectives have been explored to achieve personalization, including meta-learning (Fallah et al., 2020; Acar et al., 2021b; Jeon et al., 2024), multi-task learning (Smith et al., 2017; Dinh et al., 2022), transfer learning (Yang et al., 2019), prototype-based (Tan et al., 2022; Xu et al., 2023), and clustering (Ghosh et al., 2020; Sattler et al., 2020). Additionally, hypernetworks have also been employed to predict client model parameters (Shamsian et al., 2021; Scott et al., 2024), while regularization techniques balance personalization with global model performance (T Dinh et al., 2020; Li et al., 2020; Acar et al., 2021a; Li et al., 2021). Some strategies split the model, learning shared feature extractor and personalized classifier, or vice-versa (Arivazhagan et al., 2019; Liang et al., 2020; Collins et al., 2021). These methodologies yield point estimate solutions, which are susceptible to overfitting, particularly in scenarios where clients possess limited data.

### 2.3. Bayesian Personalized FL

Bayesian PFL leverage various Bayesian inference techniques to estimate the personalized posterior distribution. For instance, pFedBayes (Zhang et al., 2022) utilizes variational inference, while others have explored Laplace approximation (Liu et al., 2023) and Markov chain Monte Carlo (MCMC) methods (Kotelevskii et al., 2022; Bhatt et al., 2024). The formulation of global distributions in these Bayesian frameworks varies. Some methods, such as pFedBayes, employ arithmetic aggregation of personalized distribution parameters, while others approximate global distributions as the product of personalized one (Bhatt et al., 2024). In particular, pFedVEM (Zhu et al., 2023) claims the use of hierarchical Bayesian modeling, but they do not adhere to the principle due to missing hyper-priors over the parameters of personalized posteriors. Studies utilize Gaussian Processes for personalization (Yin et al., 2020; Achituve et al., 2021; Tang et al., 2022) with mixed-effect modeling to characterize feature extractors and personalized classifier layers (Kotelevskii et al., 2022) and functional space prior to enhancing collaboration (Makhija et al., 2024).

A hierarchical Bayesian model for FL is explored to define the hyper-prior as a normal-inverse-Wishart distribution over personalized posterior parameters (Kim & Hospedales, 2023). However, by fixing the personalized posterior covariance to a identity matrix, this approach reduces the local models to non-Bayesian neural networks. Similarly, a

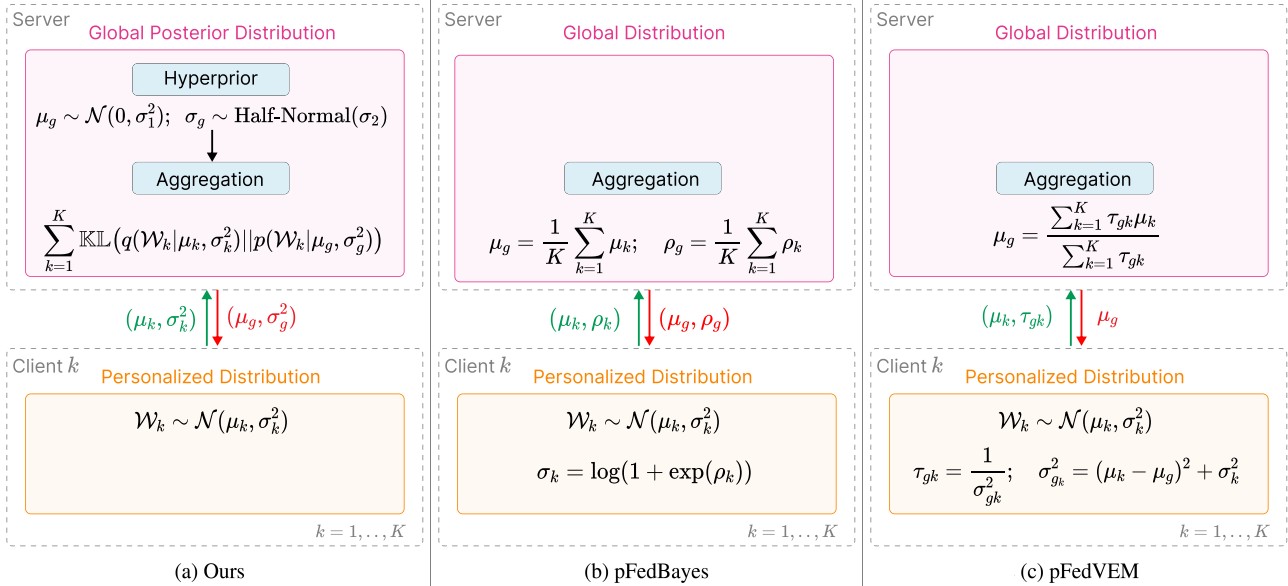

*Figure 2.* Overview of our PFL framework based on hierarchical Bayesian inference (**left**) in comparison with existing Bayesian-based PFLs such as pFedBayes (Zhang et al., 2022) (**middle**) and pFedVEM (Zhu et al., 2023) (**right**). As each client uploads its updated personalized distribution parameters $(\mu_k, \sigma_k^2)$ to the server and then downloads the aggregated global parameters $(\mu_g, \sigma_g^2)$, our framework derives a global posterior distribution $p(\mathcal{W}_k|\mu_g, \sigma_g^2)$ to aggregate the personalized posteriors by defining a hyper-prior over their parameters. Other methods heuristically specify the parameters of a global distribution as weighted sums of the personalized ones.

federated variational inference method proposes to model distribution parameters of Bayesian neural networks as latent variables (Hassan et al., 2023). Yet, they only model the weights of the first hidden layers as random variables, while the remaining weights are MAP estimates.

## 3. Hierarchical Bayesian Inference for PFL

As shown in Figure 2, our hierarchical Bayesian inference PFL framework derives a personalized posterior distribution over the parameters of each client's local model based on a Bayesian neural network (BNN). We further define a hyper-prior distribution over the parameters of these personalized posteriors on the server. This hyper-prior enables us to aggregate these personalized posteriors by computing a global posterior distribution when the clients upload their parameters to the server at each iteration. Moreover, for the joint approximate inference, we present a stochastic variational inference scheme that split the computation of the overall objective into a server part for aggregation and a client part for retaining personalization.

### 3.1. BNN-based Personalized Posterior distribution

We consider a FL system comprising a central server and $K$ decentralized clients. In this framework, each client employs a neural network $f(\cdot)$ as its local model, parame-

terized by weights $\mathcal{W}_k$, to capture the underlying structure of its local data $\mathcal{D}_k$. The likelihood is thus expressed as

$$p(\mathcal{D}_k|\mathcal{W}_k) = \prod_{(x,y)\in\mathcal{D}_k} p(y|f(x;\mathcal{W}_k)) \qquad (1)$$

We specify a multivariate Gaussian with a diagonal covariance as the prior over the weights of each client's local model $\mathcal{W}_k$ as

$$p(\mathcal{W}_k|\boldsymbol{\mu}_k, \boldsymbol{\sigma}_k^2) = \mathcal{N}(\mathcal{W}_k|\boldsymbol{\mu}_k, \boldsymbol{\sigma}_k^2 I) \qquad (2)$$

where $\boldsymbol{\mu}_k$ and $\boldsymbol{\sigma}_k^2$ denote the mean and variance of the Gaussian distribution.

### 3.2. Global Posterior Distribution Specification

We propose to set a hyper-prior distribution over the parameters of the personalized posteriors. Notably, the parameters of client $k$'s personalized posterior $\boldsymbol{\theta}_k = \{\boldsymbol{\mu}_k, \boldsymbol{\sigma}_k^2\}$ are direct instantiations of the hyper-prior $p(\boldsymbol{\theta}_g|\alpha)$, where $\alpha$ denotes hyper-parameters. Specifically, we model $p(\boldsymbol{\theta}_g|\alpha)$ as the product of a Gaussian distribution and a Half-Normal distribution (Leone et al., 1961):

$$\begin{aligned} p(\boldsymbol{\theta}_g|\alpha) &= p(\boldsymbol{\mu}_g, \boldsymbol{\sigma}_g|\alpha) \\ &= \mathcal{N}(\boldsymbol{\mu}_g|0, \sigma_1^2 I)\,\text{Half-Normal}(\boldsymbol{\sigma}_g|\sigma_2) \end{aligned} \qquad (3)$$

where $\boldsymbol{\theta}_g = (\boldsymbol{\mu}_g, \boldsymbol{\sigma}_g^2)$ and $\alpha = \{\sigma_1^2, \sigma_2\}$.

We adopt Half-Normal distribution as a prior over $\boldsymbol{\sigma}_g^2$ as it is computationally stable and efficient (Röver et al., 2021).

The global posterior distribution can thus be written as

$$p(\boldsymbol{\theta}_g, \mathcal{W}|\mathcal{D}, \alpha) \propto p(\boldsymbol{\theta}_g|\alpha) \prod_{k=1}^{K} p(\mathcal{D}_k|\mathcal{W}_k)p(\mathcal{W}_k|\boldsymbol{\theta}_g) \quad (4)$$

where $\mathcal{W} = \{\mathcal{W}_k\}$ denotes the local BNNs' weights of all the clients, and $\mathcal{D} = \{\mathcal{D}_k\}$ are all the data. The exact computation of both the global and the local posterior distributions are intractable due to the non-linearity inherent in neural networks. Therefore, we employ variational inference (VI) for approximation.

### 3.3. Stochastic VI for the Hierarchical Bayesian PFL

We specify a variational distribution $q(\boldsymbol{\theta}_g, \mathcal{W}; \phi, \boldsymbol{\Theta})$ to approximate the true posteriors as follows:

$$q(\boldsymbol{\theta}_g, \mathcal{W}; \phi, \boldsymbol{\Theta}) = q(\boldsymbol{\theta}_g; \phi) \prod_{k=1}^{K} q(\mathcal{W}_k; \boldsymbol{\theta}_k) \quad (5)$$

where, $\boldsymbol{\Theta} = \{\boldsymbol{\theta}_k\}_{k=1}^{K}$. Here, $q(\boldsymbol{\theta}_g; \phi)$ denotes the variational distribution of the global posterior parameters and $q(\mathcal{W}_k; \boldsymbol{\theta}_k)$ represents the personalized distribution over client $k$'s local model weights.

We minimize the Kullback-Leibler ($\mathbb{KL}$) divergence between the variational distribution and the true posterior with respect to the variational parameters $(\phi, \boldsymbol{\Theta})$:

$$\begin{aligned} \phi^*, \boldsymbol{\Theta}^* &= \operatorname*{argmin}_{\phi, \boldsymbol{\Theta}} \mathbb{KL}\left[q(\boldsymbol{\theta}_g, \mathcal{W}; \phi, \boldsymbol{\Theta})||p(\boldsymbol{\theta}_g, \mathcal{W}|\mathcal{D}, \alpha)\right] \\ &= \operatorname*{argmin}_{\phi, \boldsymbol{\Theta}} \sum_{k=1}^{K} \left[ E_{q(\mathcal{W}_k; \boldsymbol{\theta}_k)}\left[-\log p(\mathcal{D}_k|\mathcal{W}_k)\right] \right. \\ &\quad + E_{q(\boldsymbol{\theta}_g; \phi)}\left[\mathbb{KL}\left(q(\mathcal{W}_k; \boldsymbol{\theta}_k)||p(\mathcal{W}_k|\boldsymbol{\theta}_g)\right)\right] \right] \\ &\quad + \mathbb{KL}\left(q(\boldsymbol{\theta}_g; \phi)||p(\boldsymbol{\theta}_g|\alpha)\right) \end{aligned}$$

$$(6)$$

The resulting overall objective is known as evidence lower bound (ELBO) (Neal & Hinton, 1998; Jaakkola & Jordan, 2000; Blundell et al., 2015). The first term facilitates the adaptation of the local models to the corresponding local data. The second term aggregates the personalized posterior distributions through the global posterior distribution. The last term acts as a regularization for the global posterior parameters. This ELBO thus effectively balances personalization with global model generalization. We further split the computation of the overall objective into a server part for aggregation and a client part for retaining personalization to avoid the potential communication overhead caused by frequently synchronizing personalized posterior parameters.

**Server Part of the Objective:** On the central server, we aggregate the parameters of the personalized posteriors

through the global posterior by optimizing the second and the third terms of the overall objective in Equation (6) with respect to $\phi$:

$$\begin{aligned} \mathcal{L}_g &= \sum_{k=1}^{K} E_{q(\boldsymbol{\theta}_g; \phi)}\left[\mathbb{KL}\left(q(\mathcal{W}_k; \boldsymbol{\theta}_k)||p(\mathcal{W}_k|\boldsymbol{\theta}_g)\right)\right] \\ &\quad + \mathbb{KL}\left(q(\boldsymbol{\theta}_g; \phi)||p(\boldsymbol{\theta}_g|\alpha)\right) \end{aligned} \quad (7)$$

For communication efficiency, we specify $q(\boldsymbol{\theta}_g; \phi)$ as a Dirac distribution centered on a specific value $\hat{\boldsymbol{\theta}}_g$ as $q(\boldsymbol{\theta}_g; \phi) = \delta(\boldsymbol{\theta}_g - \hat{\boldsymbol{\theta}}_g)$. This leads to a simplified server objective:

$$\mathcal{L}_g = \sum_{k=1}^{K} \mathbb{KL}\left(q(\mathcal{W}_k; \boldsymbol{\theta}_k)||p(\mathcal{W}_k|\hat{\boldsymbol{\theta}}_g)\right) - \log p\left(\hat{\boldsymbol{\theta}}_g|\alpha\right)$$

$$(8)$$

By incorporating the hyper-prior in Equation (3), we have:

$$\begin{aligned} \mathcal{L}_g &= \sum_{k=1}^{K} \mathbb{KL}\left(q(\mathcal{W}_k; \boldsymbol{\theta}_k)||p(\mathcal{W}_k|\hat{\boldsymbol{\theta}}_g)\right) + \\ &\quad \lambda_1 ||\hat{\boldsymbol{\mu}}_g||^2 + \lambda_2 ||\hat{\boldsymbol{\sigma}}_g||^2 \end{aligned} \quad (9)$$

where $||\hat{\boldsymbol{\mu}}_g||^2 = \sum_{i=1}^{M} \hat{\boldsymbol{\mu}}_{i,g}^2$, $||\hat{\boldsymbol{\sigma}}_g||^2 = \sum_{i=1}^{M} \hat{\boldsymbol{\sigma}}_{i,g}^2$ and $M = |\mathcal{W}_k|$. $\lambda_1 \propto \frac{1}{\sigma_1^2}$ and $\lambda_2 \propto \frac{1}{\sigma_2^2}$ are regularization coefficients.

**Client Part of the Objective:** In each client, we update the personalized posterior by minimizing its expected negative log-likelihood over the local data while regularizing it with the global posterior. Thus, the $k$-th client part of the overall objective in Equation (6) is :

$$\begin{aligned} \mathcal{L}_k &= \sum_{k=1}^{K} \left[ E_{q(\mathcal{W}_k; \boldsymbol{\theta}_k)}\left[-\log p(\mathcal{D}_k|\mathcal{W}_k)\right] \right. \\ &\quad + \mathbb{KL}\left(q(\mathcal{W}_k; \boldsymbol{\theta}_k)||p(\mathcal{W}_k|\hat{\boldsymbol{\theta}}_g)\right) \right] \end{aligned} \quad (10)$$

Specifically, we specify each client's local model as a personalized BNN. Thus its variational distribution is:

$$\begin{aligned} q(\mathcal{W}_k; \boldsymbol{\theta}_k) &= q(\mathcal{W}_k; \boldsymbol{\mu}_k, \boldsymbol{\sigma}_k^2) \\ &= \prod_{i=1}^{M} \mathcal{N}(\mathcal{W}_{i,k}|\boldsymbol{\mu}_{i,k}, \boldsymbol{\sigma}_{i,k}^2) \end{aligned} \quad (11)$$

where $\boldsymbol{\mu}_k = \{\boldsymbol{\mu}_{i,k}\}$ and $\boldsymbol{\sigma}_k^2 = \{\boldsymbol{\sigma}_{i,k}^2\}$. $\boldsymbol{\mu}_{i,k}$ and $\boldsymbol{\sigma}_{i,k}^2$ denote the mean and variance of a Gaussian distribution for the $i$-th weight of $k$-th client's local. To ensure that $\boldsymbol{\sigma}_{i,k}$ takes non-negative values, we replace each $\boldsymbol{\sigma}_{i,k}$ by another parameters $\boldsymbol{\rho}_{i,k}$ such that $\boldsymbol{\sigma}_{i,k} = \log(1 + \exp(\boldsymbol{\rho}_{i,k}))$.

In the context of FL systems, where each client data remains inaccessible to the central server, we employ a block-coordinate optimization approach (Wu & Lange, 2008; Diakonikolas & Orecchia, 2018) to optimize the objective

functions. The optimization process consists of two alternating steps: first on the server side, we optimize the global variational parameters while keeping the client-specific variational parameters $\Theta$ fixed; second, on the client side, we update the personalized variational parameters $\Theta$ for multiple rounds while fixing the $\phi$ as a constant.

## 4. Theoretical Analysis

We provide theoretical analysis for the statistical convergence of the personalized posterior distributions, and show the existing Bayesian PFL methods are special instances of our framework.

### 4.1. Statistical Convergence

We analyze the convergence behavior of the personalized distributions for each client $k$: $p(\mathcal{W}_k|\boldsymbol{\theta}_k)$ towards the global mean $\bar{\boldsymbol{\mu}} = E[\boldsymbol{\mu}_g|\mathcal{W}]$, which is close to the pooled estimate $\bar{\mathcal{W}}$ by the FedAvg method.

**Theorem 4.1.** *Let the hyperparameters $\alpha$ to be fixed, and approximate the global posterior with $p(\boldsymbol{\mu}_g, \boldsymbol{\sigma}_g^2|\mathcal{W}) = \delta(\boldsymbol{\mu}_g - \bar{\boldsymbol{\mu}})\delta(\boldsymbol{\sigma}_g^2 - \bar{\boldsymbol{\sigma}}^2)$ for simplicity, then the convergence of the personalized distribution mean:*

$$E[\boldsymbol{\theta}_k|\mathcal{W}] = \beta_k\bar{\boldsymbol{\mu}} + (1 - \beta_k)\hat{\boldsymbol{\theta}}_k \qquad (12)$$

*where $\beta_k = \frac{\boldsymbol{\sigma}_k^2}{\boldsymbol{\sigma}_k^2 + \bar{\boldsymbol{\sigma}}^2}$.*

It indicates that the mean of the personalized distribution parameters: $E[\boldsymbol{\theta}_k|\mathcal{W}]$ lies in between the local MLE estimate $\hat{\boldsymbol{\theta}}_k = \bar{\mathcal{W}}_k$ and the global mean $\bar{\boldsymbol{\mu}}$. The convergence of the personalized parameters towards the global mean is thus governed by $\beta_k$. There is larger convergence for clients with smaller measurement precision (e.g., due to smaller data sizes or noisy labels), since $\beta_k \to 1$ as $\boldsymbol{\sigma}_k^2 \to \infty$.

### 4.2. Special Instances of our Hierarchical Bayesian Inference PFL

Our proposed framework is fully general for heuristic aggregations of existing Bayesian PFL methods. We can recover the existing methods' objectives as special cases by introducing additional constraints to the server-side objective in Equation (9). We first expand Equation (9) as:

$$\mathcal{L}_g = \sum_{k=1}^{K} E_{q(\mathcal{W}_k; \boldsymbol{\theta}_k)}\left[-\log p(\mathcal{W}_k|\hat{\boldsymbol{\theta}}_g)\right] \\ \lambda_1||\hat{\boldsymbol{\mu}}_g||^2 + \lambda_2||\hat{\boldsymbol{\sigma}}_g||^2 \qquad (13)$$

**Example I: pFedBayes (Zhang et al., 2022).** Given the constraints that the regularization coefficients are to zero

(i.e., $\lambda_1 = \lambda_2 = 0$), Equation 13 is simplifies to:

$$\mathcal{L}_{g'} = \sum_{k=1}^{K} E_{q(\mathcal{W}_k; \boldsymbol{\theta}_k)}\left[-\log p(\mathcal{W}_k|\hat{\boldsymbol{\theta}}_g)\right] \qquad (14)$$

By setting the derivative of $\mathcal{L}_{g'}$ with respect to $\hat{\boldsymbol{\theta}}_g$ to 0, we obtain pFedBayes' aggregation rule for $\hat{\boldsymbol{\mu}}_{g'}$:

$$\hat{\boldsymbol{\mu}}_{g'} = \frac{1}{K}\sum_{k=1}^{K}\boldsymbol{\mu}_k; \qquad \hat{\boldsymbol{\sigma}}_{g'}^2 = \frac{1}{K}\sum_{k=1}^{K}[(\boldsymbol{\mu}_k - \hat{\boldsymbol{\mu}}_{g'})^2 + \boldsymbol{\sigma}_k^2]$$

Furthermore, given an additional constraint $\boldsymbol{\mu}_k = \hat{\boldsymbol{\mu}}_{g'}$, we recover pFedBayes' aggregation rule for $\hat{\boldsymbol{\sigma}}_{g'}^2$:

$$\hat{\boldsymbol{\sigma}}_{g'}^2 = \frac{1}{K}\sum_{k=1}^{K}\boldsymbol{\sigma}_k^2$$

**Example II: pFedVEM (Zhu et al., 2023).** With the same constraint that $\lambda_1 = \lambda_2 = 0$ and assuming each client equipped with a specific global distribution variance $\hat{\boldsymbol{\sigma}}_{g_k''}^2$, we can re-write Equation 13 as:

$$\mathcal{L}_{g''} = \sum_{k=1}^{K} E_{q(\mathcal{W}_k; \boldsymbol{\theta}_k)}\left[-\log p(\mathcal{W}_k|\hat{\boldsymbol{\theta}}_{g_k''})\right] \qquad (15)$$

where $\hat{\boldsymbol{\theta}}_{g_k''} = \{\hat{\boldsymbol{\mu}}_{g''}, \hat{\boldsymbol{\sigma}}_{g_k''}^2\}$

By setting the derivative of $\mathcal{L}_{g''}$ with respect to $\hat{\boldsymbol{\theta}}_{g''}$ to 0, we have pFedVEM's aggregation rule for $\hat{\boldsymbol{\mu}}_{g''}$:

$$\hat{\boldsymbol{\mu}}_{g''} = \frac{\sum_{k=1}^{K}\hat{\boldsymbol{\tau}}_{g_k''}\boldsymbol{\mu}_k}{\sum_{k=1}^{K}\hat{\boldsymbol{\tau}}_{g_k''}}$$

where $\hat{\boldsymbol{\tau}}_{g_k''} = \frac{1}{\hat{\boldsymbol{\sigma}}_{g_k''}^2}$ and $\hat{\boldsymbol{\sigma}}_{g_k''}^2 = (\boldsymbol{\mu}_k - \hat{\boldsymbol{\mu}}_{g''})^2 + \boldsymbol{\sigma}_k^2$.

As shown above, the arithmetic aggregation adopted by the existing Bayesian PFL methods compute the global distribution by simply averaging the personalized posterior parameters. The samples from the averaged global distribution limits their ability to capture non-IID characteristics of the local models, blurring the statistical heterogeneity of the local data. In contrast, the personalized posterior parameters in our proposed framework are sampled from the convex combination of the averaged distribution and each personalized distribution's local estimates, as in Theorem 4.1. In Section 6.3, our ablation study shows that the arithmetic aggregation scheme setting $\lambda_1 = \lambda_2 = 0$ leads to inferior performance.

## 5. Algorithm Implementation

We provide the pseudo-code of our algorithm along with a description in Appendix A.3.

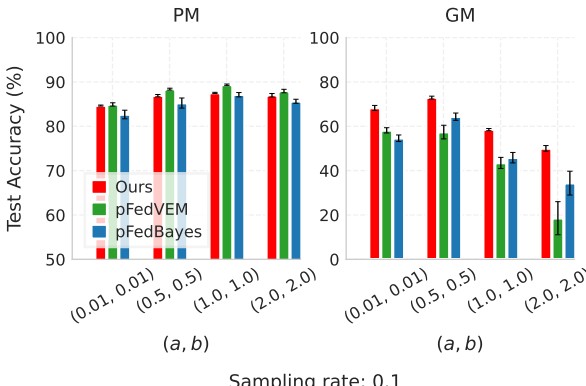
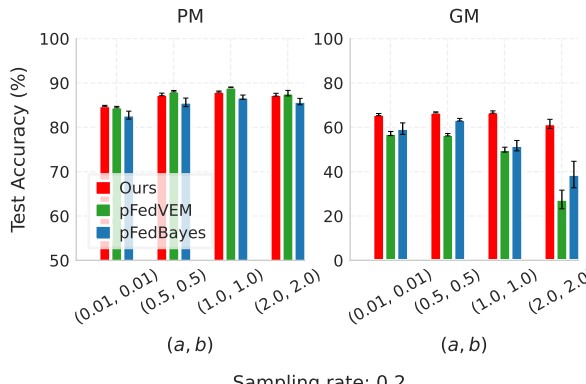

*Figure 3.* Average test accuracy (%) of Personalized Model (PM) and Global Model (GM) across varying statistical heterogeneity controlled by $(a, b)$ and sampling rate (i.e., 0.1 and 0.2). Although the PM performance are comparable, our GM consistently outperforms other methods across varying levels of statistical heterogeneity and under different client participation rates.

**Local BNN Optimization:** To compute the local BNN for each client, as outlined in Equation (10), we utilize recent advancements as Bayes-by-Backprop (Blundell et al., 2015), which formulates an unbiased estimator for gradients through re-parameterization.

**Monte Carlo Approximation:** We employ Monte Carlo estimation to approximate the expectation term in Equation (10). The estimator is expressed as follows:

$$E_{q(\mathcal{W}_k; \boldsymbol{\theta}_k)} \left[ -\log p(\mathcal{D}_k | \mathcal{W}_k) \right] \approx -\frac{1}{S} \sum_{s=1}^{S} \log p\left( \mathcal{D}_k | \mathcal{W}_k^{(i)} \right)$$

where we draw $S$ samples from the personalized posterior distribution $q(\mathcal{W}_k; \boldsymbol{\theta}_k)$.

**Base-Head Architecture:** Recent studies have highlighted the effectiveness of Base-Head architecture in PFL scenarios (Collins et al., 2021; Kotelevskii et al., 2022; Zhu et al., 2023). Following this paradigm, the global model is decomposed into base and head components. The corresponding distribution parameters $\hat{\boldsymbol{\theta}}_g$ of the global model are also partitioned into base ($\hat{\boldsymbol{\theta}}_{g;base}$) and head ($\hat{\boldsymbol{\theta}}_{g;head}$) components, such that $\hat{\boldsymbol{\theta}}_g = \hat{\boldsymbol{\theta}}_{g;base} \oplus \hat{\boldsymbol{\theta}}_{g;head}$, where $\oplus$ denotes a concatenation operation. Similarly, for each client $k$, the local model is decomposed into base and head components, with the corresponding personalized posterior distribution parameters expressed as: $\boldsymbol{\theta}_k = \boldsymbol{\theta}_{k;base} \oplus \boldsymbol{\theta}_{k;head}$. The base component facilitates the extraction of common feature representations across clients, while the head component enables client-specific personalization.

Following (T Dinh et al., 2020; Zhu et al., 2023), at each communication round, clients perform local Bayesian inference, deriving personalized posterior distribution over local

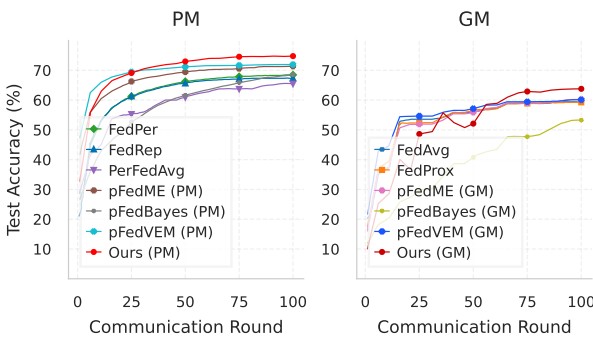

*Figure 4.* Test accuracy vs. communication rounds for 100 clients on CIFAR10. Our framework shows rapid convergence.

model parameters, regularized by global posterior distribution. A subset of clients communicates these personalized posterior distribution parameters to the central server. The server aggregates these personalized posteriors through a global posterior distribution. Subsequently, the resulting global posterior parameters are broadcast to all clients, facilitating the next training iteration. The code is available online [1].

## 6. Experiments

### 6.1. Synthetic Experiment

In this experiment, we focus on assessing the performance of the global model across diverse statistical heterogeneity scenarios of our proposed framework. We adopt (Li

---

[1] https://github.com/mahendrathapa/pFedHB

*Table 1.* Average test accuracy of PMs and test accuracy of GM on FMNIST, CIFAR10, and CIFAR100 datasets for 50, 100, and 200 clients. Test accuracies are presented in percentage and the best results are highlighted in bold.

| Dataset | Method | 50 Clients | | 100 Clients | | 200 Clients | |
| --- | --- | --- | --- | --- | --- | --- | --- |
| | | PM (%) | GM (%) | PM (%) | GM (%) | PM (%) | GM (%) |
| FMNIST | Local BNN only | $88.7 \pm 0.9$ | – | $86.9 \pm 0.9$ | – | $85.2 \pm 0.4$ | – |
| | FedAvg | – | $83.5 \pm 0.4$ | – | $85.4 \pm 0.3$ | – | $85.9 \pm 0.2$ |
| | FedProx | – | $84.8 \pm 0.5$ | – | $\mathbf{86.3 \pm 0.2}$ | – | $\mathbf{86.5 \pm 0.1}$ |
| | Scaffold | – | $85.6 \pm 0.2$ | – | $85.4 \pm 0.1$ | – | $84.6 \pm 0.0$ |
| | FedPer | $91.4 \pm 0.1$ | – | $90.7 \pm 0.1$ | – | $89.7 \pm 0.1$ | – |
| | FedRep | $91.5 \pm 0.1$ | – | $90.7 \pm 0.1$ | – | $89.9 \pm 0.1$ | – |
| | PerFedAvg | $88.7 \pm 0.2$ | – | $88.6 \pm 0.1$ | – | $88.3 \pm 0.2$ | – |
| | pFedME | $91.9 \pm 0.1$ | $82.0 \pm 0.7$ | $91.4 \pm 0.1$ | $84.4 \pm 0.6$ | $90.6 \pm 0.1$ | $85.1 \pm 0.1$ |
| | pFedBayes | $91.9 \pm 0.1$ | $83.5 \pm 0.3$ | $91.3 \pm 0.1$ | $84.2 \pm 0.3$ | $90.5 \pm 0.1$ | $84.4 \pm 0.1$ |
| | pFedVEM | $91.8 \pm 0.1$ | $83.9 \pm 0.3$ | $91.4 \pm 0.1$ | $85.6 \pm 0.2$ | $90.7 \pm 0.1$ | $86.2 \pm 0.2$ |
| | Ours | $\mathbf{92.2 \pm 0.3}$ | $\mathbf{86.1 \pm 0.4}$ | $\mathbf{91.6 \pm 0.3}$ | $85.9 \pm 0.2$ | $\mathbf{90.8 \pm 0.1}$ | $85.2 \pm 0.1$ |
| CIFAR10 | Local BNN only | $54.9 \pm 1.1$ | – | $50.0 \pm 0.1$ | – | $45.6 \pm 0.2$ | – |
| | FedAvg | – | $57.7 \pm 0.9$ | – | $59.4 \pm 0.6$ | – | $59.2 \pm 0.3$ |
| | FedProx | – | $58.0 \pm 0.7$ | – | $59.4 \pm 0.5$ | – | $59.1 \pm 0.2$ |
| | Scaffold | – | $60.4 \pm 0.3$ | – | $59.8 \pm 0.2$ | – | $55.4 \pm 0.3$ |
| | FedPer | $72.7 \pm 0.3$ | – | $68.4 \pm 0.4$ | – | $63.4 \pm 0.3$ | – |
| | FedRep | $71.4 \pm 0.3$ | – | $67.4 \pm 0.4$ | – | $62.8 \pm 0.2$ | – |
| | PerFedAvg | $62.9 \pm 0.8$ | – | $65.6 \pm 0.8$ | – | $64.2 \pm 0.1$ | – |
| | pFedME | $72.3 \pm 0.1$ | $56.6 \pm 1.0$ | $71.4 \pm 0.2$ | $60.1 \pm 0.3$ | $68.5 \pm 0.2$ | $58.7 \pm 0.2$ |
| | pFedBayes | $71.4 \pm 0.3$ | $52.0 \pm 1.0$ | $68.5 \pm 0.3$ | $53.2 \pm 0.7$ | $64.6 \pm 0.2$ | $51.4 \pm 0.3$ |
| | pFedVEM | $73.2 \pm 0.2$ | $56.0 \pm 0.4$ | $71.9 \pm 0.1$ | $60.1 \pm 0.2$ | $70.1 \pm 0.3$ | $59.4 \pm 0.3$ |
| | Ours | $\mathbf{77.1 \pm 0.1}$ | $\mathbf{65.9 \pm 0.9}$ | $\mathbf{74.7 \pm 0.3}$ | $\mathbf{63.8 \pm 0.3}$ | $\mathbf{70.4 \pm 0.2}$ | $\mathbf{59.5 \pm 0.7}$ |
| CIFAR100 | Local BNN only | $39.5 \pm 1.3$ | – | $31.5 \pm 1.0$ | – | $26.2 \pm 0.4$ | – |
| | FedAvg | – | $51.7 \pm 0.5$ | – | $49.4 \pm 0.7$ | – | $44.7 \pm 0.5$ |
| | FedProx | – | $48.4 \pm 0.6$ | – | $45.5 \pm 0.5$ | – | $42.4 \pm 0.3$ |
| | Scaffold | – | $47.2 \pm 0.4$ | – | $41.4 \pm 0.7$ | – | $30.0 \pm 0.1$ |
| | FedPer | $49.7 \pm 0.7$ | – | $39.3 \pm 0.7$ | – | $30.6 \pm 0.9$ | – |
| | FedRep | $50.9 \pm 0.9$ | – | $41.2 \pm 0.6$ | – | $30.5 \pm 0.6$ | – |
| | PerFedAvg | $52.1 \pm 0.4$ | – | $48.3 \pm 0.5$ | – | $40.1 \pm 0.3$ | – |
| | pFedME | $52.5 \pm 0.5$ | $47.9 \pm 0.5$ | $47.6 \pm 0.5$ | $45.1 \pm 0.3$ | $41.6 \pm 1.8$ | $41.5 \pm 1.6$ |
| | pFedBayes | $49.6 \pm 0.3$ | $42.5 \pm 0.5$ | $46.5 \pm 0.2$ | $41.3 \pm 0.3$ | $40.1 \pm 0.3$ | $37.4 \pm 0.3$ |
| | pFedVEM | $61.0 \pm 0.4$ | $52.8 \pm 0.4$ | $56.2 \pm 0.4$ | $52.3 \pm 0.4$ | $51.1 \pm 0.6$ | $49.2 \pm 0.5$ |
| | Ours | $\mathbf{66.6 \pm 0.4}$ | $\mathbf{56.0 \pm 0.3}$ | $\mathbf{62.2 \pm 0.3}$ | $\mathbf{55.4 \pm 0.1}$ | $\mathbf{56.4 \pm 0.6}$ | $\mathbf{52.4 \pm 0.4}$ |

et al., 2020) to generate synthetic data that utilize parameters $a$ and $b$ to modulate statistical heterogeneity across clients. Specifically, $a$ controls the degree of divergence between local models, while $b$ regulates the variation in local data distributions among clients. The dataset is designed to train a 10 class classifier on 60 dimensional real-value data. Following (Li et al., 2020), we distribute the data to $N = 50$ clients according to a power law distribution. We run the experiments for 100 communication rounds. To simulate real-world federated learning conditions with partial client participation, we measure performance across sampling rates of 0.1 and 0.2. For all methods, we employ a Bayesian Fully Connected network (Appendix A.4) for both global and local models, with a local learning rate of 0.005, local batch size same as local data size, and 10 local number of epochs. We tune the hyperparameters of pFedBayes and pFedVEM around their recommended values.

We evaluate the performance of both Personalized Models (PMs) and Global Model (GM). The experimental results in Figure 3 demonstrate that our framework consistently outperforms the competing methods in GM performance across varying levels of statistical heterogeneity and under different client participation rates. Notably other methods' GMs exhibit significant performance degradation under high heterogeneity conditions. This indicates that the computation of a higher level posterior distribution over the parameters of the personalized posterior distribution on the server effectively harness the heterogeneous statistical strength of the clients' non-IID data to construct a robust and stable global model. The performance of PMs remains comparable across all approaches. This may be attributed to the relative simplicity of the synthetic dataset, characterized by low feature dimensionality and a small data size, which enables all methods to effectively capture the underlying non-IID characteristics.

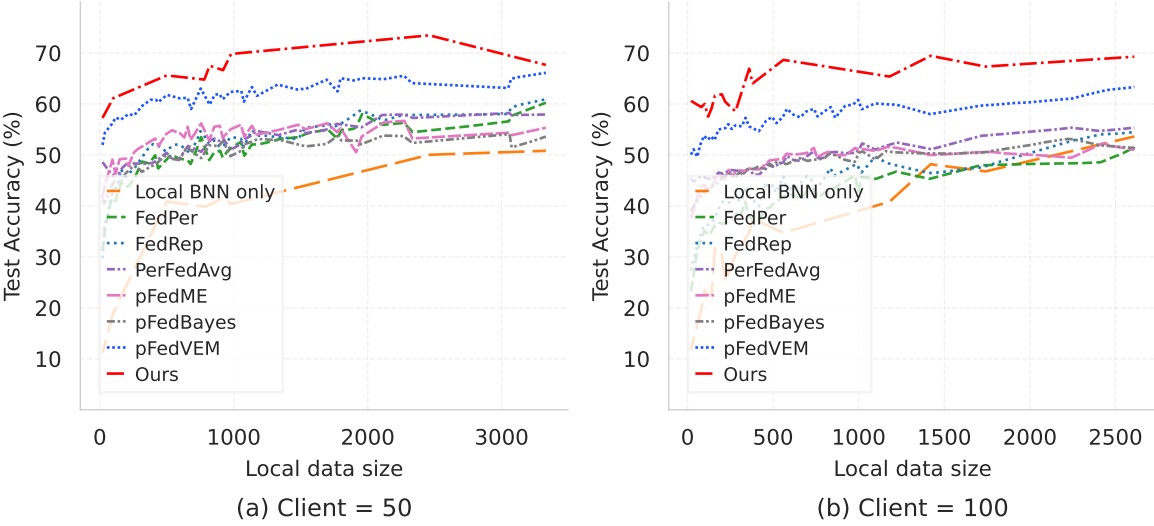

*Figure 5.* Test accuracy vs. local data size for 50 and 100 clients on CIFAR100. Our framework provides greater collaborative learning benefits for clients.

### 6.2. Performance Comparison on Real Data

We examine various categories of statistical heterogeneity that exist in real-world FL scenarios – *label distribution skew*: clients possess varying label distributions, *label concept drift*: feature distributions differ among clients, and *data quantity disparity*: clients have different amounts of data (Ye et al., 2023; Zhu et al., 2023). Following (Zhu et al., 2023), we assess performance on Fashion-MNIST (FMNIST) (Xiao et al., 2017), CIFAR10 (Krizhevsky et al., 2009), and CIFAR100 (Krizhevsky et al., 2009) datasets. For label distribution skew, each client on FMNIST and CIFAR10 datasets has unique local data of 5 labels out of 10. For label concept drift in CIFAR100, the task is superclass predictions. We compare the performance of our approach with the following FL frameworks: FedAvg (McMahan et al., 2017), FedProx (Li et al., 2020), Scaffold (Karimireddy et al., 2020), FedPer (Arivazhagan et al., 2019), FedRep (Collins et al., 2021), PerFedAvg (Fallah et al., 2020), pFedME (T Dinh et al., 2020), pFedBayes (Zhang et al., 2022), pFedVEM (Zhu et al., 2023), and Local BNN only, which refers to BNN optimized independently at each client.

We run the experiments for 100 communication rounds. To emulate real-world FL conditions characterized by partial client participation, each client has a probability of 0.1 to communicate its local model parameters back to the server during each round of communication following a binomial distribution. The experimental setup encompasses varying numbers of clients, specifically 50, 100, and 200. For model architectures, we employ Bayesian MLP with one hidden layer for FMNIST, a LeNet-like Bayesian CNN for Cifar10, and a 6-layer Bayesian CNN for CIFAR-100 similar to the

configuration in (Zhu et al., 2023) (Appendix A.4). For each dataset, we conduct grid search for hyperparameter optimization for our proposed method using $K = 100$ client and apply those configurations to $K = 50$ and $K = 200$ case. We report the average test accuracy (%) with corresponding standard error, computed over three independent runs, each with a different random seed. We conduct all the experiments using NVIDIA A5000 GPUs.

#### 6.2.1. PERFORMANCE COMPARISON

Table 1 presents a comprehensive performance across three datasets: FMNIST, CIFAR10 and CIFAR100, with varying number of clients (50, 100, 200). Our Global Model (GM) consistently outperforms existing approaches. On the CIFAR10 dataset, our GM outperforms competing models by 5.5%, 3.7% for 50 and 100 clients, respectively, and achieves comparable performance for 200 clients. Our GM's performance on the CIFAR100 dataset is consistently superior, surpassing other models by 3.2%, 3.1%, and 3.2% for 50, 100, and 200 clients, respectively. These results indicate that our framework effectively shares the heterogeneous statistical strength across local models via a hierarchical setup. For the FMNIST dataset, especially with 100 and 200 clients, our GM maintains performance comparable to state-of-the-art approaches, likely due to the relative simplicity of the dataset and the limited presence of non-IID characteristics. Our Personalized Model (PM) demonstrates superior performance compared to the competing methods. On the FMNIST dataset, our approach exhibits superior performance across all client configurations. On the CIFAR10 dataset, our PM consistently outperforms competing algo-

*Table 2.* Average test accuracy of PM and GM on CIFAR100 dataset of our method using different global model regularization coefficients. Test accuracies are presented in percentage and the best results are highlighted in bold.

| Regularization Coefficients | 50 Clients | | 100 Clients | | 200 Clients | |
|---|---|---|---|---|---|---|
| $(\lambda_1, \lambda_2)$ | PM (%) | GM (%) | PM (%) | GM (%) | PM (%) | GM (%) |
| (0, 0) | $61.9 \pm 1.1$ | $53.1 \pm 0.2$ | $59.4 \pm 0.4$ | $52.9 \pm 0.2$ | $55.3 \pm 0.2$ | $51.1 \pm 0.5$ |
| (1, 1) | $63.2 \pm 1.1$ | $54.7 \pm 0.4$ | $60.7 \pm 0.3$ | $54.4 \pm 0.4$ | $56.2 \pm 0.3$ | $51.8 \pm 0.4$ |
| (0, 5) | $61.7 \pm 0.6$ | $53.8 \pm 0.2$ | $59.6 \pm 0.4$ | $53.8 \pm 0.2$ | $55.9 \pm 0.1$ | $52.4 \pm 0.7$ |
| (5, 0) | $65.4 \pm 1.2$ | $54.6 \pm 0.4$ | $60.7 \pm 0.3$ | $52.9 \pm 0.3$ | $53.9 \pm 1.1$ | $48.1 \pm 0.8$ |
| (5, 5) | $\mathbf{66.6 \pm 0.4}$ | $\mathbf{56.0 \pm 0.3}$ | $\mathbf{62.2 \pm 0.3}$ | $\mathbf{55.4 \pm 0.1}$ | $\mathbf{56.4 \pm 0.6}$ | $\mathbf{52.4 \pm 0.4}$ |
| (10, 10) | $\mathbf{66.6 \pm 1.3}$ | $51.9 \pm 0.7$ | $\mathbf{62.2 \pm 0.2}$ | $50.1 \pm 3.7$ | $55.4 \pm 0.4$ | $49.2 \pm 1.6$ |

rithms, with performance gains of 3.9%, 2.8%, and 0.3% for 50, 100, and 200 clients, respectively. The performance gains are even more pronounced on the CIFAR100 dataset, where our PM surpasses competing algorithms by substantial margins of 5.6%, 6.0%, and 5.3% for 50, 100, and 200 clients, respectively. This demonstrates the ability of our method to effectively retain distinctive characteristics of local data via enabling client-level model personalization.

Moreover, Figure 4 shows the convergence behavior of various algorithms on CIFAR10 for 100 clients. Our framework demonstrates rapid convergence, achieving competitive performance at approximately half the total communication rounds required by the baseline methods. This efficiency is crucial in FL, where minimizing communication rounds is important. Similar convergence patterns are observed for scenarios involving 50 and 200 clients (Appendix A.7).

Furthermore, Table 1 shows the overall performance of PFL methods consistently surpasses that of Local BNN only approach across all experimental settings. We further investigate the collaborative learning benefits across clients with diverse local data sizes. Specifically, we perform an experiment using CIFAR100 across 50 and 100 clients and plot test accuracy over local data size in Figure 5. It shows that our framework provides greater benefits to clients compared to the competing approaches and incentivize participation from clients for the collaborative training process.

### 6.3. Ablation Study

We conduct an ablation study to investigate the impact of regularization on model performance. For this, we perform experimentation using CIFAR100 across 50, 100, and 200 clients. Table 2 shows the model's performance without regularization $(\lambda_1, \lambda_2) = (0, 0)$ consistently underperforms compared to regularized configurations. It suggests that hyper-prior over personalized posterior parameters plays an important role in enhancing both PM and GM generalization. The best performance is achieved with regularization coefficients $(\lambda_1, \lambda_2) = (5, 5)$. Importantly, with either $\lambda_1$ or $\lambda_2$ set to 0, the model's performance degrades. This

indicates that the hyper-prior over both $\hat{\boldsymbol{\mu}}_g$ and $\hat{\boldsymbol{\sigma}}_g$ plays a crucial role in enhancing the model performance. Furthermore, excessive regularization (e.g., (10, 10)) negatively impacts the model performance, particularly for the global model. This suggests that overly stringent regularization can potentially constrain the model's learning capacity.

We investigate the impact of updating order of local model's base and head components in Appendix A.5. Consistent with (Collins et al., 2021; Zhu et al., 2023), our findings indicate that updating the head component prior to the base yields the best performance. We also demonstrate the scalability of our proposed method to a large number of clients and the generalization to more complex datasets, along with the evaluation of uncertainty quantification in Appendix A.9.

## 7. Conclusion

Our hierarchical Bayesian inference PFL framework specifies a hyper-prior over the parameters of personalized posteriors. This enables us to jointly compute a global posterior for aggregation and local posteriors for personalization. Results shows by sharing heterogeneous statistical strength across clients while retaining their distinctive characteristics, our framework yields state-of-the-art performance. Our hierarchical Bayesian approach for PFL offers a principled framework that enables effective generalization in low-data regimes, quantifies uncertainty, and enhances robustness (Cao et al., 2023).

## Acknowledgements

This work is supported by the National Science Foundation under NSF award No.2045804 and National Institutes of Health under NIH award No.1R35GM156653-01. We acknowledge the Research Computing at Rochester Institute of Technology (RIT, 2024) for the computational resources.

## Impact Statement

Federated learning enables collaborative model training across decentralized client devices while preserving data privacy, and demonstrates significant potential in various real-world applications. Our framework effectively addresses the challenges that federated learning systems face in general due to statistical heterogeneity and limited data availability on individual clients.

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

# A. Appendix

## A.1. Stochastic VI for the Hierarchical Bayesian PFL

We minimize the Kullback-Leibler ($\mathbb{KL}$) divergence between the variational distribution $q(\boldsymbol{\theta}_g, \mathcal{W}; \phi, \boldsymbol{\Theta})$ and true posterior $p(\boldsymbol{\theta}_g, \mathcal{W}|\mathcal{D}, \alpha)$ with respect to the variational parameters $(\phi, \boldsymbol{\Theta})$:

$$\phi^*, \boldsymbol{\Theta}^* = \operatorname*{argmin}_{\phi, \boldsymbol{\Theta}} \mathbb{KL}\left[q(\boldsymbol{\theta}_g, \mathcal{W}; \phi, \boldsymbol{\Theta})||p(\boldsymbol{\theta}_g, \mathcal{W}|\mathcal{D}, \alpha)\right]$$

By the definition of $\mathbb{KL}$ divergence:

$$\mathbb{KL}\left[q(\boldsymbol{\theta}_g, \mathcal{W}; \phi, \boldsymbol{\Theta})||p(\boldsymbol{\theta}_g, \mathcal{W}|\mathcal{D}, \alpha)\right] = E_{q(\boldsymbol{\theta}_g, \mathcal{W};\phi,\boldsymbol{\Theta})}\left[\log \frac{q(\boldsymbol{\theta}_g, \mathcal{W}; \phi, \boldsymbol{\Theta})}{p(\boldsymbol{\theta}_g, \mathcal{W}|\mathcal{D}, \alpha)}\right]$$

Using Bayes' theorem:

$$p(\boldsymbol{\theta}_g, \mathcal{W}|\mathcal{D}, \alpha) = \frac{p(\mathcal{D}|\mathcal{W})p(\mathcal{W}|\boldsymbol{\theta}_g)p(\boldsymbol{\theta}_g|\alpha)}{p(\mathcal{D}|\alpha)}$$

Thus, the $\mathbb{KL}$ divergence becomes:

$$E_{q(\boldsymbol{\theta}_g, \mathcal{W};\phi,\boldsymbol{\Theta})}\left[\log q(\boldsymbol{\theta}_g, \mathcal{W}; \phi, \boldsymbol{\Theta}) - \log p(\mathcal{D}|\mathcal{W}) - \log p(\mathcal{W}|\boldsymbol{\theta}_g) - \log p(\boldsymbol{\theta}_g|\alpha) + \log p(\mathcal{D}|\alpha)\right]$$

Since $p(\mathcal{D}|\alpha)$ is independent of $\phi$ and $\boldsymbol{\Theta}$, minimizing the $\mathbb{KL}$ divergence is equivalent to minimizing the following ELBO:

$$\mathcal{L} = E_{q(\boldsymbol{\theta}_g, \mathcal{W};\phi,\boldsymbol{\Theta})}\left[-\log p(\mathcal{D}|\mathcal{W}) - \log p(\mathcal{W}|\boldsymbol{\theta}_g) - \log p(\boldsymbol{\theta}_g|\alpha) + \log q(\boldsymbol{\theta}_g, \mathcal{W}; \phi, \boldsymbol{\Theta})\right]$$

We specify a variational distribution $q(\boldsymbol{\theta}_g, \mathcal{W}; \phi, \boldsymbol{\Theta})$ to approximate the true posteriors as follows:

$$\log q(\boldsymbol{\theta}_g, \mathcal{W}; \phi, \boldsymbol{\Theta}) = \log q(\boldsymbol{\theta}_g; \phi) + \log q(\mathcal{W}; \boldsymbol{\Theta})$$

where $q(\boldsymbol{\theta}_g; \phi)$ denotes the variational distribution of the global posterior parameters and $q(\mathcal{W}; \boldsymbol{\Theta})$ represents the personalized posterior distribution over client local model weights.

Thus, the ELBO becomes:

$$\mathcal{L} = E_{q(\boldsymbol{\theta}_g, \mathcal{W};\phi,\boldsymbol{\Theta})}\left[-\log p(\mathcal{D}|\mathcal{W}) - \log p(\mathcal{W}|\boldsymbol{\theta}_g) - \log p(\boldsymbol{\theta}_g|\alpha) + \log q(\boldsymbol{\theta}_g; \phi) + \log q(\mathcal{W}; \boldsymbol{\Theta})\right]$$

$$= E_{q(\boldsymbol{\theta}_g, \mathcal{W};\phi,\boldsymbol{\Theta})}\left[-\log p(\mathcal{D}|\mathcal{W}) + \log \frac{q(\mathcal{W}; \boldsymbol{\Theta})}{p(\mathcal{W}|\boldsymbol{\theta}_g)} + \log \frac{q(\boldsymbol{\theta}_g; \phi)}{p(\boldsymbol{\theta}_g|\alpha)}\right]$$

As clients are conditionally independent given the common prior $\boldsymbol{\theta}_g$, the ELBO becomes:

$$\mathcal{L} = \sum_{k=1}^{K}\left[E_{q(\mathcal{W}_k;\boldsymbol{\theta}_k)}\left[-\log p(\mathcal{D}_k|\mathcal{W}_k)\right] + E_{q(\boldsymbol{\theta}_g;\phi)}[E_{q(\mathcal{W}_k;\boldsymbol{\theta}_k)}[\log \frac{q(\mathcal{W}_k;\boldsymbol{\theta}_k)}{p(\mathcal{W}_k|\boldsymbol{\theta}_g)}]]\right] + E_{q(\boldsymbol{\theta}_g;\phi)}\left[\log \frac{q(\boldsymbol{\theta}_g;\phi)}{p(\boldsymbol{\theta}_g|\alpha)}\right]$$

$$= \sum_{k=1}^{K}\left[E_{q(\mathcal{W}_k;\boldsymbol{\theta}_k)}\left[-\log p(\mathcal{D}_k|\mathcal{W}_k)\right] + E_{q(\boldsymbol{\theta}_g;\phi)}\left[\mathbb{KL}\left(q(\mathcal{W}_k;\boldsymbol{\theta}_k)||p(\mathcal{W}_k|\boldsymbol{\theta}_g)\right)\right]\right] + \mathbb{KL}\left(q(\boldsymbol{\theta}_g;\phi)||p(\boldsymbol{\theta}_g|\alpha)\right)$$

## A.2. Statistical Convergence

After computing the approximate global posterior in Equation (4) via stochastic variational inference (i.e., Section 3.3), we analyze the convergence of the personalized distribution for each client $k$: $p(\mathcal{W}_k|\boldsymbol{\theta}_k)$, where $\boldsymbol{\theta}_k = \{\boldsymbol{\mu}_k, \boldsymbol{\sigma}_k^2\}$ are the parameters of the client $k$'s personalized posterior, as in Section 3.2, and $\mathcal{W}_k$ denotes the weights of the local model (i.e., Bayesian neural networks).

Specifically, we analyze the convergence behavior of the personalized distributions towards the global mean $\bar{\boldsymbol{\mu}} = E[\boldsymbol{\mu}_g|\mathcal{W}]$, which is close to the pooled estimate $\bar{\mathcal{W}}$. Let the hyperparameters $\alpha$ to be fixed, and we adopt the approximation to the

global posterior $p(\boldsymbol{\mu}_g, \boldsymbol{\sigma}_g^2 | \mathcal{W}) = \delta(\boldsymbol{\mu}_g - \bar{\boldsymbol{\mu}})\delta(\boldsymbol{\sigma}_g^2 - \bar{\boldsymbol{\sigma}}^2)$ for simplicity, then the marginal distribution of the personalized distribution parameters is:

$$p(\boldsymbol{\theta}_k | \mathcal{W}) = \int p(\boldsymbol{\theta}_k | \mathcal{W}_k, \boldsymbol{\theta}_g) p(\boldsymbol{\theta}_g | \mathcal{W}) d\boldsymbol{\theta}_g \tag{16}$$

$$\approx p(\boldsymbol{\theta_k} | \mathcal{W}_k, \bar{\boldsymbol{\mu}}, \bar{\boldsymbol{\sigma}}^2) \tag{17}$$

note that $\boldsymbol{\theta}_g = \{\boldsymbol{\mu}_g, \boldsymbol{\sigma}_g^2\}$.

Meanwhile, the above marginal posterior can be expressed as

$$p(\boldsymbol{\theta}_k | \mathcal{W}_k, \bar{\boldsymbol{\mu}}, \bar{\boldsymbol{\sigma}}^2) \propto p(\mathcal{W}_k | \boldsymbol{\theta}_k) \cdot p(\boldsymbol{\theta}_k | \bar{\boldsymbol{\mu}}, \bar{\boldsymbol{\sigma}}^2) \tag{18}$$

$$= \mathcal{N}(\mathcal{W}_k | \boldsymbol{\theta}_k) \cdot \mathcal{N}(\boldsymbol{\theta}_k | \bar{\boldsymbol{\mu}}, \bar{\boldsymbol{\sigma}}^2) \tag{19}$$

$$= \exp(-\frac{(\hat{\boldsymbol{\theta}}_k - \boldsymbol{\theta}_k)^2}{\boldsymbol{\sigma}_k^2}) \cdot \exp(-\frac{(\boldsymbol{\theta}_k - \bar{\boldsymbol{\mu}})^2}{\bar{\boldsymbol{\sigma}}^2}) \tag{20}$$

$$= \exp(-(\frac{1}{2\boldsymbol{\sigma}_k^2} + \frac{1}{2\bar{\boldsymbol{\sigma}}^2})\boldsymbol{\theta}_k^2 + (\frac{\hat{\boldsymbol{\theta}}_k}{\boldsymbol{\sigma}_k^2} + \frac{\bar{\boldsymbol{\mu}}}{\bar{\boldsymbol{\sigma}}^2})\boldsymbol{\theta}_k - (\frac{\hat{\boldsymbol{\theta}}_k^2}{2\boldsymbol{\sigma}_k^2} + \frac{\bar{\boldsymbol{\mu}}^2}{2\bar{\boldsymbol{\sigma}}^2})) \tag{21}$$

where $\hat{\boldsymbol{\theta}}_k = \bar{\mathcal{W}}_k$ denotes the local MLE estimate.

Since the marginal posterior $p(\boldsymbol{\theta}_k | \mathcal{W}_k, \bar{\boldsymbol{\mu}}, \bar{\boldsymbol{\sigma}}^2) = \mathcal{N}(\boldsymbol{\theta}_k | E[\boldsymbol{\theta}_k | \mathcal{W}], Var[\boldsymbol{\theta}_k | \mathcal{W}])$ is a Gaussian due to conjugacy, by matching the coefficients in the quadratic and linear terms of the two expressions, we have the variance of the marginal distribution as

$$\frac{1}{2Var[\boldsymbol{\theta}_k | \mathcal{W}]} = \frac{1}{2\boldsymbol{\sigma}_k^2} + \frac{1}{2\bar{\boldsymbol{\sigma}}^2} \tag{22}$$

$$Var[\boldsymbol{\theta}_k | \mathcal{W}] = \frac{\boldsymbol{\sigma}_k^2 \bar{\boldsymbol{\sigma}}^2}{\boldsymbol{\sigma}_k^2 + \bar{\boldsymbol{\sigma}}^2} \tag{23}$$

and the mean of the marginal distribution as

$$\frac{E[\boldsymbol{\theta}_k | \mathcal{W}]}{Var[\boldsymbol{\theta}_k | \mathcal{W}]} = \frac{\hat{\boldsymbol{\theta}}_k}{\boldsymbol{\sigma}_k^2} + \frac{\bar{\boldsymbol{\mu}}}{\bar{\boldsymbol{\sigma}}^2} \tag{24}$$

$$E[\boldsymbol{\theta}_k | \mathcal{W}] = (\frac{\hat{\boldsymbol{\theta}}_k}{\boldsymbol{\sigma}_k} + \frac{\bar{\boldsymbol{\mu}}^2}{\bar{\boldsymbol{\sigma}}^2}) \cdot Var[\boldsymbol{\theta}_k | \mathcal{W}] \tag{25}$$

$$= \beta_k \bar{\boldsymbol{\mu}} + (1 - \beta_k)\hat{\boldsymbol{\theta}}_k \tag{26}$$

where $\beta_k = \frac{\boldsymbol{\sigma}_k^2}{\boldsymbol{\sigma}_k^2 + \bar{\boldsymbol{\sigma}}^2}$.

It indicates that the mean of the personalized distribution parameters: $E[\boldsymbol{\theta}_k | \mathcal{W}]$ lies in between the local MLE estimate $\hat{\boldsymbol{\theta}}_k = \bar{\mathcal{W}}_k$ and the global mean $\bar{\boldsymbol{\mu}}$. The convergence of the personalized parameters towards the global mean is governed by $\beta_k$. Thus, we see that there is larger convergence for clients with smaller measurement precision (e.g., due to smaller data sizes or noisy labels), since $\beta_k \to 1$ as $\boldsymbol{\sigma}_k^2 \to \infty$.

### A.3. Algorithmic Description

In our proposed framework, clients perform local Bayesian inference at each communication round, deriving personalized posterior distribution over local model parameters. A subset of clients communicates these parameters to the central server. The central server then aggregates these personalized posteriors to construct a global posterior, which is regularized by a hyper-prior. Subsequently, the server broadcasts the latest global posterior to all clients for the next training iteration. The full procedure of our proposed works is shown in Algorithm 1.

### A.4. Model Architecture

The Model architectures for Synthetic, FMNIST, CIFAR10, and CIFAR100 datasets used in this proposed work are shown in Figure 6. We employ a Bayesian Fully Connected network for synthetic data, a Bayesian MLP with one hidden layer

for FMNIST, a LeNet-like Bayesian CNN for Cifar10, and a 6-layer Bayesian CNN for CIFAR-100. Each architecture incorporates ReLU activation functions following Bayesian Fully Connected networks and Bayesian Convolutional Layer. All models are structurally decomposed into Base and Head components, with the Head component comprising only the last layer, while the Base component encompasses all preceding layers. Notably, as the synthetic data model consists of Bayesian Fully Connected layers only, it doesn't have a Base component.

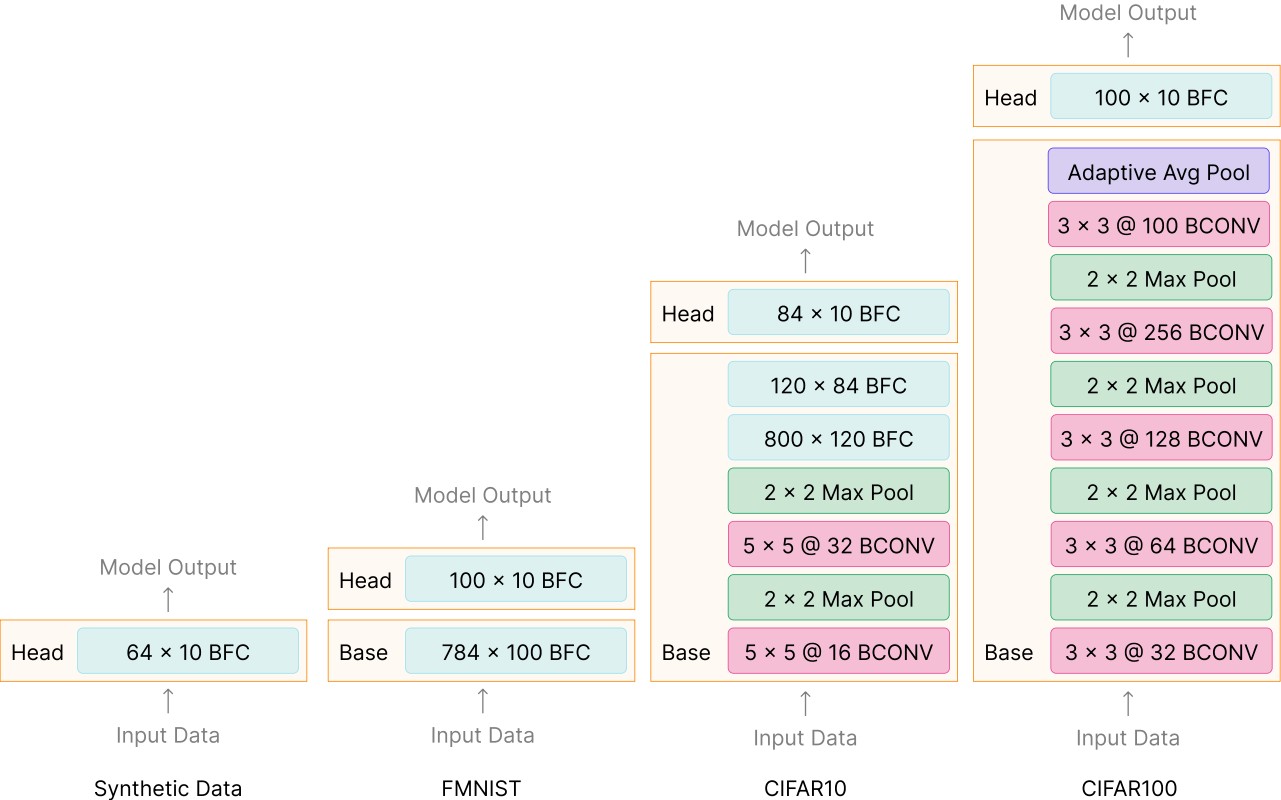

*Figure 6.* Model architectures for Synthetic, FMNIST, CIFAR10 and CIFAR100 datasets. BFC refers to Bayesian Fully Connected Layer and BCONV refers to Bayesian Convolutional Layer.

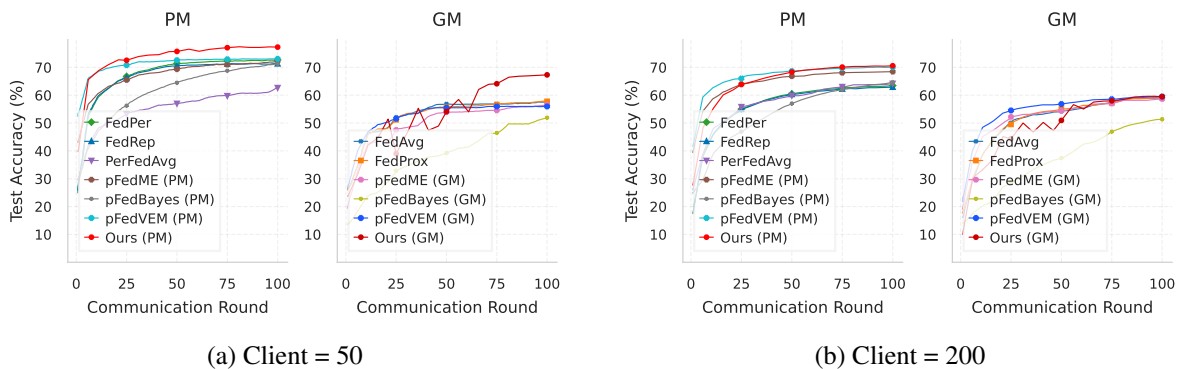

*Figure 7.* Test accuracy vs. communication rounds for 50 and 100 clients on CIFAR10.

## A.5. Base-Head Updating Order

We investigate the impact of different updating orders of the local model's base and head components. Specifically, we consider three updating orders: updating the base component only (Oh et al., 2022), updating the base followed by the head,

*Table 3.* Average test accuracy of PM and GM on CIFAR10 dataset of our method using different update orders. Test accuracies are presented in percentage and the best results are highlighted in bold.

| Update Order | Sharing Component | 50 Clients | | 100 Clients | | 200 Clients | |
|---|---|---|---|---|---|---|---|
| | | PM (%) | GM (%) | PM (%) | GM (%) | PM (%) | GM (%) |
| Base only | Base only | $76.8 \pm 0.2$ | – | $72.9 \pm 0.2$ | – | $68.3 \pm 0.4$ | – |
| Base $\rightarrow$ Head | Both base & head | $76.7 \pm 0.4$ | $65.1 \pm 0.5$ | $72.6 \pm 0.8$ | $60.6 \pm 0.6$ | $68.3 \pm 0.4$ | $57.3 \pm 0.6$ |
| Head $\rightarrow$ Base | Both base & head | $\mathbf{77.1 \pm 0.1}$ | $\mathbf{65.9 \pm 0.9}$ | $\mathbf{74.7 \pm 0.3}$ | $\mathbf{63.8 \pm 0.3}$ | $\mathbf{70.4 \pm 0.2}$ | $\mathbf{59.5 \pm 0.7}$ |

and updating the head followed by the base (Collins et al., 2021; Zhu et al., 2023). We perform experimentation using CIFAR10 across 50, 100 and 200 clients. The experimental results in Table 3 indicate that the strategy of updating the head followed by the base consistently yields superior performance. This updating order facilitates initial adaptation of the head components to the client's local data, which is subsequently followed by refinement of shared feature representation through updates to the base components.

### A.6. Hyperparameters

We list all the hyperparameters of our method for the Synthetic, FMNIST, CIFAR10, and CIFAR100 datasets in Table 4. Our initial hyperparameters configuration is based on the recommendation provided by pFedVEM (Zhu et al., 2023). Subsequently, we leverage grid search to finetune these hyperparameters for our proposed method. Compared to baseline methods, our proposed approach incurs additional computational overhead of tuning the regularization coefficients $\lambda_1$ and $\lambda_2$ via grid search. Our ablation study, as in Section 6.3, demonstrate that these regularization coefficients are critical to the superior performance for Bayesian personalized federated learning (pFL) in general. Disabling them leads to a substantial degradation in performance, highlighting the major performance bottleneck of the prior approaches.

*Table 4.* Hyperparameters of our proposed method for various datasets. For the synthetic data, which consists only of the head component, base component hyperparameters are not applicable (denoted by –).

| Hyperparameter | Synthetic Data | FMNIST | CIFAR10 | CIFAR100 |
|---|---|---|---|---|
| Local Batch size for base | – | 50 | 50 | 50 |
| Local Batch size for head | Local data size | Local data size | Local data size | Local data size |
| Local optimizer | Adam | Adam | Adam | Adam |
| Local learning rate for base | – | 0.001 | 0.001 | 0.001 |
| Local learning rate for head | 0.005 | 0.001 | 0.001 | 0.001 |
| Local epochs for base | – | 2 | 2 | 4 |
| Local epochs for head | 10 | 10 | 10 | 20 |
| Global learning rate for base | – | 0.01 | 0.01 | 0.01 |
| Global learning rate for head | 0.01 | 0.01 | 0.01 | 0.01 |
| Global epochs for base | – | 5 | 5 | 5 |
| Global epochs for head | 10 | 5 | 5 | 5 |
| Global optimizer | Adam | Adam | Adam | Adam |
| $\lambda_1$ | 1 | 5 | 5 | 5 |
| $\lambda_2$ | 1 | 5 | 5 | 5 |
| Local $\mathbb{KL}$ coefficient for base | – | $10^{-6}$ | $10^{-6}$ | $10^{-6}$ |
| Local $\mathbb{KL}$ coefficient for head | 1 | 1 | 1 | 1 |

### A.7. Convergence Behavior

Figure 7 show the convergence behavior of various algorithms on the CIFAR10 dataset for 50 and 200 clients, respectively. Our proposed method demonstrates rapid convergence for 50 clients, achieving the competitive performance of PM and GM at approximately one-fourth and half of the communication rounds needed by the baseline approaches. In the case of 200 clients, our method's convergence rate is comparable to other competitive approaches.

## A.8. Computational Cost

We measure the average local training time per communication round across three configurations: non-Bayesian (McMahan et al., 2017), non-Bayesian base with Bayesian head (Zhu et al., 2023), and full-Bayesian approaches (Zhang et al., 2022). Our proposed method also employs a full-Bayesian framework. For this, we perform an experiment using CIFAR10, with all 100 clients participating in each communication round. As shown in Table 5, full-Bayesian methods incur high computation costs, leading to a longer local training time per communication round.

*Table 5.* Average local training time (in seconds) per communication round on CIFAR10 with all 100 clients available

| Method | Local Training Time (seconds) |
| --- | --- |
| non-Bayesian | $21.42 \pm 1.36$ |
| non-Bayesian base with Bayesian head | $43.14 \pm 3.22$ |
| full-Bayesian | $90.38 \pm 4.50$ |

## A.9. Additional Experiments

### A.9.1. SCALABILITY UNDER LARGE NUMBER OF CLIENTS

To evaluate the scalability of our proposed methods under a significantly large number of clients, we conduct experiments on the CIFAR10 dataset with 1000 clients, where each client has unique local data of 5 labels out of 10. As shown in Table 6, our methods scale well with a large number of clients.

*Table 6.* Average test accuracy of PM and GM on CIFAR10 dataset for 1000 clients. Test accuracies are presented in percentage and the best results are highlighted in bold.

| Method | PM (%) | GM (%) |
| --- | --- | --- |
| FedAvg | – | $48.7 \pm 1.1$ |
| pFedVEM | $59.0 \pm 0.4$ | $48.6 \pm 0.9$ |
| Ours | $\mathbf{61.0 \pm 0.3}$ | $\mathbf{52.3 \pm 0.4}$ |

### A.9.2. GENERALIZATION UNDER COMPLEX DATASET

To assess the generalization capability of our proposed method on a more complex dataset with a large number of categories, we conduct experiments on the Tiny-ImageNet dataset (Le & Yang, 2015). The experimental setup includes 50 clients, each with local data consisting of 50 labels out of 200. As presented in Table 7, our proposed framework achieves the best performance.

*Table 7.* Average test accuracy of PM and GM on Tiny-ImageNet dataset for 50 clients. Test accuracies are presented in percentage and the best results are highlighted in bold.

| Method | PM (%) | GM (%) |
| --- | --- | --- |
| FedAvg | – | $10.1 \pm 0.7$ |
| pFedVEM | $33.5 \pm 0.2$ | $18.5 \pm 0.4$ |
| Ours | $\mathbf{42.4 \pm 0.1}$ | $\mathbf{25.7 \pm 0.1}$ |

### A.9.3. UNCERTAINTY QUANTIFICATION

To demonstrate the effectiveness of the Bayesian framework in uncertainty quantification, we evaluate predictive uncertainty on the CIFAR10 dataset. We adopt (Hahn et al., 2022) to simulate a noisy labels scenario by applying symmetric flipping (Van Rooyen et al., 2015) with a noise ratio of 0.2. We report the test Expected Calibration Error (ECE) (Naeini et al., 2015) and test accuracy. As shown in Table 8, our proposed method's uncertainty quantification is comparable with pFedVEM in the Personalized Model setting, while consistently achieving higher accuracy across all experimental configurations.

*Table 8.* Average test Expected Calibration Error (ECE) and test accuracy (Acc) of PM and GM on CIFAR10 dataset under noisy labels. Test accuracies are presented in percentage and the best results are highlighted in bold.

| Method | 50 Clients | | 100 Clients | |
|---|---|---|---|---|
| | PM: ECE (Acc) | GM: ECE (Acc) | PM: ECE (Acc) | GM: ECE (Acc) |
| FedAvg | – | $0.37 \pm 0.06$ ($36.8 \pm 0.9$) | – | $0.35 \pm 0.08$ ($37.9 \pm 2.5$) |
| pFedVEM | $\mathbf{0.13 \pm 0.01}$ ($70.1 \pm 0.4$) | $\mathbf{0.19 \pm 0.01}$ ($46.1 \pm 2.3$) | $0.17 \pm 0.00$ ($68.0 \pm 0.4$) | $\mathbf{0.22 \pm 0.00}$ ($49.6 \pm 2.9$) |
| Ours | $0.15 \pm 0.00$ ($\mathbf{72.6 \pm 0.4}$) | $0.28 \pm 0.00$ ($\mathbf{59.8 \pm 0.3}$) | $\mathbf{0.16 \pm 0.01}$ ($\mathbf{70.3 \pm 0.3}$) | $0.26 \pm 0.00$ ($\mathbf{57.6 \pm 0.7}$) |

### A.9.4. COMPARISON WITH NON-FL SETUP

We compare the performance of our proposed method against a non-federated learning (non-FL) centralized setup, implemented using a deterministic neural network trained on the entire dataset. As shown in Table 9, the performance of the centralized setup does not necessarily serve as an upper bound for the federated learning (FL) setup, particularly in the case of personalized model. This may be attributed to the centralized model's need to generalize across the entire data distribution, which often leads to a reliance on globally shared features while neglecting local patterns that may be critical for certain subsets of the data. In contrast, personalized model captures local patterns effectively, resulting in improved performance.

*Table 9.* Average test accuracy of PMs and GM for 50 clients for Federated Learning (FL) setup and test accuracy for non-FL Centralized Learning setup on FMNIST, CIFAR10, and CIFAR100 datasets. Test accuracies are presented in percentage. NN refers to determinstic neural networks.

| Method | FMNIST | | CIFAR10 | | CIFAR100 | |
|---|---|---|---|---|---|---|
| | PM (%) | GM (%) | PM (%) | GM (%) | PM (%) | GM (%) |
| FL (Ours; 50 Clients) | $92.2 \pm 0.3$ | $86.1 \pm 0.4$ | $77.1 \pm 0.1$ | $65.9 \pm 0.9$ | $66.6 \pm 0.4$ | $56.0 \pm 0.3$ |
| Non-FL (Centralized; NN) | $88.6 \pm 0.2$ | | $68.3 \pm 0.4$ | | $57.3 \pm 0.5$ | |

---

**Algorithm 1** Our Method

---

**Input:**

$K$: Total number of clients indexed by $k$

$T$: Total number of communication rounds

$C$: Client sampling rate

$E_b$: Total number of local base epochs

$E_h$: Total number of local head epochs

$E_s$: Total number of global epochs

**Initialize:**

Global BNN model characterized by Global posterior distribution parameters $\hat{\boldsymbol{\theta}}_g^0$

Local BNN model characterized by Personalized posterior distribution for all client $\{\boldsymbol{\theta}_k^0\}\ \forall k \in K$

**for** $t = 0$ **to** $T - 1$ **do**

 **Server executes:**

 **for** $k = 1$ **to** $K$ **in parallel do**

  $\boldsymbol{\theta}_k^{t+1} \leftarrow$ **ClientUpdate**$(k, \hat{\boldsymbol{\theta}}_g^t)$

 $\mathcal{S}^t \leftarrow$ Select a random subset of clients using Binomial distribution $B(K, C)$

 $\hat{\boldsymbol{\theta}}_g^{t+1} \leftarrow$ **ServerUpdate**$(\{\boldsymbol{\theta}_{k'}^{t+1}\}_{k' \in \mathcal{S}^t})$

**ServerUpdate**$(\{\boldsymbol{\theta}_{k'}^{t+1}\}_{k' \in \mathcal{S}^t})$**:**

 ▷ Optimize global BNN model's base component distribution parameters using Equation (9)

 **for** $i = 1$ **to** $E_s$ **do**

  $\hat{\boldsymbol{\theta}}_{g;base}^{t+1} = \underset{\hat{\boldsymbol{\theta}}_{g;base}}{\text{argmin}}\ \mathcal{L}_g(\hat{\boldsymbol{\theta}}_{g;base}^t, \hat{\boldsymbol{\theta}}_{g;head}^t, \{\boldsymbol{\theta}_{k'}^{t+1}\}_{k' \in \mathcal{S}^t})$

 ▷ Optimize global BNN model's head component distribution parameters using Equation (9)

 **for** $i = 1$ **to** $E_s$ **do**

  $\hat{\boldsymbol{\theta}}_{g;head}^{t+1} = \underset{\hat{\boldsymbol{\theta}}_{g;head}}{\text{argmin}}\ \mathcal{L}_g(\hat{\boldsymbol{\theta}}_{g;base}^{t+1}, \hat{\boldsymbol{\theta}}_{g;head}^t, \{\boldsymbol{\theta}_{k'}^{t+1}\}_{k' \in \mathcal{S}^t})$

 $\hat{\boldsymbol{\theta}}_g^{t+1} \leftarrow \hat{\boldsymbol{\theta}}_{g;base}^{t+1} \oplus \hat{\boldsymbol{\theta}}_{g;head}^{t+1}$

 return $\hat{\boldsymbol{\theta}}_g^{t+1}$

**ClientUpdate**$(k, \hat{\boldsymbol{\theta}}_g^t)$**:**

 ▷ Initialize local BNN model's base component distribution parameters

 $\boldsymbol{\theta}_{k;base}^t \leftarrow \hat{\boldsymbol{\theta}}_{g;base}^t$

 ▷ Optimize local BNN model's head component distribution parameters using Equation (10)

 **for** $i = 1$ **to** $E_h$ **do**

  $\boldsymbol{\theta}_{k;head}^{t+1} = \underset{\boldsymbol{\theta}_{k;head}}{\text{argmin}}\ \mathcal{L}_k(\boldsymbol{\theta}_{k;base}^t, \boldsymbol{\theta}_{k;head}^t, \hat{\boldsymbol{\theta}}_{g;head}^t)$

 ▷ Optimize local BNN model's base component distribution parameters using Equation (10)

 **for** $i = 1$ **to** $E_b$ **do**

  $\boldsymbol{\theta}_{k;base}^{t+1} = \underset{\boldsymbol{\theta}_{k;base}}{\text{argmin}}\ \mathcal{L}_k(\boldsymbol{\theta}_{k;base}^t, \boldsymbol{\theta}_{k;head}^{t+1}, \hat{\boldsymbol{\theta}}_{g;base}^t)$

 $\boldsymbol{\theta}_k^{t+1} \leftarrow \boldsymbol{\theta}_{k;base}^{t+1} \oplus \boldsymbol{\theta}_{k;head}^{t+1}$

 return $\boldsymbol{\theta}_k^{t+1}$

---

