# OpenReview forum: "Harnessing Heterogeneous Statistical Strength for Personalized Federated Learning via Hierarchical Bayesian Inference"
_ICML.cc/2025/Conference — ICML 2025 poster_

### Official Review · Reviewer_kx6d · 2025-03-08

**Overall Recommendation:** 2

**Summary:**

This paper proposes a hierarchical Bayesian inference framework for personalized federated learning (PFL) to address issues arising from statistical heterogeneity across client data. The main contribution involves specifying a conjugate hyper-prior over personalized posterior parameters, enabling joint inference of personalized and global posterior distributions. Experimental results across synthetic and real datasets (FMNIST, CIFAR10, CIFAR100).  The authors further demonstrate analyctically that existing Bayesian PFL methods (e.g., *pFedBayes* and *pFedVEM*).

**Claims And Evidence:**

The claims of achieving superior performance and addressing statistical heterogeneity are well-supported by empirical evaluations on synthetic and standard benchmark datasets (FMNIST, CIFAR10, CIFAR100). However, it is not entirely clear why the addition of global hyperpriors significantly outperforms simpler aggregation methods such as that from pFedBayes. Since the aggregation of existing methods correspond to particular priors under certain assumptions, it is unclear how an uninformative hyperprior would yield substantial performance improvements.

**Essential References Not Discussed:**

I think these two papers can do the model (or model type) that this paper describes. And I think the authors should really try and readdress their novelty in terms of these two papers. At the very least, these papers should be referenced and discussed within:

- Kim, Minyoung, and Timothy Hospedales. "Fedhb: Hierarchical bayesian federated learning." arXiv preprint arXiv:2305.04979 (2023).
- Hassan, Conor, Robert Salomone, and Kerrie Mengersen. "Federated variational inference methods for structured latent variable models." arXiv preprint arXiv:2302.03314 (2023).

**Experimental Designs Or Analyses:**

The experimental designs and analyses appear sound. The use of different client counts (50, 100, and 200) and varying data distributions to simulate realistic federated settings was well-justified. No significant issues were found in the experimental setups.

**Methods And Evaluation Criteria:**

The methods and the evaluation criteria are appropriate for assessing performance in personalized federated learning scenarios. The use of benchmark datasets (FMNIST, CIFAR10, CIFAR100) and synthetic data for controlled heterogeneity simulations aligns well with the targeted problems.

**Other Comments Or Suggestions:**

**Other comments or suggestions:** Clarifying the computational complexity of the algorithm, compared to existing methods, would strengthen the work.

**Other Strengths And Weaknesses:**

**Strengths:**
- Hierarchical Bayesian approaches are elegant and address statistical heterogeneity effectively.
- Comprehensive empirical studies clearly demonstrate the benefits and effectiveness of the method.
- The method is well-motivated, and the connections to existing methods are clearly explained.

**Weaknesses:**

- Novelty might be perceived as limited since many works propose models and algorithms for federated learning of hierarchical models nowadays.
- Computational complexity and scalability aspects are not explicitly addressed.
- The additional computational cost of optimizing global hyperpriors is not clearly justified, particularly compared to simpler arithmetic aggregation methods (e.g., pFedBayes).
- Without highly informative priors, it is unclear how substantial performance improvements are achieved.

**Questions For Authors:**

- Could you elaborate on the computational complexity of the hierarchical Bayesian method compared to existing methods?
- Could you elaborate on the novelties of your model and algorithm formulation relative to the two references that I included?
- Are there specific scenarios or practical constraints where the proposed hierarchical Bayesian approach might underperform compared to simpler methods?
- Could you clarify how much performance improvement is directly attributable to the hierarchical hyperpriors versus the optimization method itself?

**Relation To Broader Scientific Literature:**

The paper clearly situates itself within the broader scientific literature by discussing connections and distinctions to related Bayesian-based FL methods and personalized FL approaches. However, essentially this method appears primarily to introduce a prior over local parameters, a concept already capable with existing algorithms and methodlogies in the literature, though maybe not quite *explored* in directly the same manner (see references below).  Additionally, while placing a hierarchical prior might be conceptually principled, it introduces significant computational overhead compared to simpler methods like pFedBayes, which typically avoid full stochastic VI both locally and globally within each communication round. This additional computational cost and its justification are not adequately discussed.

**Theoretical Claims:**

I checked the claim regarding the special cases (the derivation showing that pFedBayes and pFedVEM methods are a special case of the proposed hierarchical Bayesian formulation).

---

> ### Author Rebuttal · Authors · 2025-03-31
>
> We thank Reviewer kx6d for taking the time to provide detailed and constructive review on our paper.
>
> **"..it is not entirely clear why the addition of global hyperpriors significantly outperforms simpler aggregation methods "**: We show that simpler aggregation methods implicitly set the regularization coefficients $\lambda_1$ and $\lambda_2$ to $0$. As in Table 2 (Page 8), our ablation study shows that $\lambda_1=\lambda_2=0$ leads to inferior performance. Simpler aggregation averages out the local posterior parameters. This hinders their ability of capturing non-IID features since the arithmetic means blur the statistical heterogeneity of the local data. Defining a hyperprior over the personalized parameters lets us share their statistical strength through the hyperprior and retaining the unique properties of local models.
>
> **"...placing a hierarchical prior...introduces significant computational overhead compared to simpler methods like pFedBayes...its justification are not adequately discussed."**: As described in Section 3.3, we split the overall ELBO into server part and client part to avoid the potential communication overhead caused by frequently synchronizing client parameters, since the client part of the ELBO can be optimized locally in multiple rounds after receiving updates from the server. Comparing with simpler methods, the only additional computation overhead, as discussed in Appendix A.4, is about tuning the regularization coefficients $\lambda_1$ and $\lambda_2$ via grid search. This can be justified by the statistically significant performance gain across benchmarks.
>
> **"Novelty might be perceived as limited...in terms of these two papers"**: Although paper (Kim, et al. 2023) proposes to define Normal-inverse Wishard (NIW) distribution and mixture models as hyper-priors over personalized posterior parameters, in the implementation, they simplify the NIW to "spiky Gaussian" by fixing the covariance to a singular identity matrix, as in their Equation (16). This means they fail to aggregate the covariances of the personalized posteriors. This is also verified by the poorer performance the paper reported in their experiments. In comparison, we propose half-Gaussian, a special case of NIW, over the covariances and effectively aggregate the personalized posteriors. Besides our more comprehensive experiments, we also provide theoretical analysis shedding light on the relationships and limitations of the existing Bayesian pFL methods. Paper (Hassan, et al. 2023) introduces a FL model with hierarchical latent variables, instead of directly performing Bayesian inference on distribution parameters as in our paper. The work's empirical evaluation is quite limited, since the model is not tested on any benchmark datasets.
>
> Although the two papers are arXiv drafts, we will discuss them in our camera-ready version and clarify the novelty of our work.
>
> **"Computational complexity and scalability aspects are not explicitly addressed"**:
> - Regarding the computational complexity, Figure 3 (Page 7) shows our method outperforms the baselines with fewer communication rounds. The average local training time (seconds) per communication round in Cifar10 with all 100 clients available is
> |Methods|Time
> |--|--
> |non-Bayes (FedAvg)|21.42
> |non-Bayes Base, Bayes Head (pFedVEM)|43.14
> |full-Bayes (Ours, pFedBayes)|90.38
>
> - For scalability, we validated and tested our framework with the largest number of clients (i.e., 200) in comparison with what the baseline methods reported on the image datasets. For example, pFedBayes uses maximum 20 clients, FedAvg 100 clients, and pFedVEM 200 clients. To demonstrate the scalability, we conduct an experiment with a total of 1000 clients on Cifar10, where each client has unique local data of 5 labels out of 10, as below:
> |Methods| PM | GM
> |--|--|--|
> |FedAvg | -| 48.7±1.1
> |pFedVEM| 59.0±0.4|48.6±0.9
> |Ours| **61.0± 0.3**| **52.3±0.4**
>
> Our method scales well with large number of clients.
>
> **"Are there specific scenarios or practical constraints where the proposed hierarchical Bayesian approach might underperform compared to simpler methods?":** Our PMs' performance is comparable with the baselines on the synthetic data. This is because the synthetic data are relatively simple in terms of lower feature dimensionality and smaller dataset sizes. The baselines can thus perform well capturing the non-IID information. The similar case is on FMNIST dataset which is relatively simple and does not have many non-IID features. Therefore, the advantages of personalization algorithms are not obvious. This can be verified by the fact that all the baselines including FedAvg achieves similar performance.
>
> **"Could you clarify how much performance improvement is directly attributable to the hierarchical hyperpriors versus the optimization method itself?"**: Our ablation study in Table 2 (Page 8) suggests that properly setting the parameters of the hyperprior is important to the performance gain.

---

> > ### Comment · Reviewer_kx6d · 2025-04-06
> >
> > Thanks for your rebuttal.
> >
> > > "...placing a hierarchical prior...introduces significant computational overhead compared to simpler methods like pFedBayes...its justification are not adequately discussed.": As described in Section 3.3, we split the overall ELBO into server part and client part to avoid the potential communication overhead caused by frequently synchronizing client parameters, since the client part of the ELBO can be optimized locally in multiple rounds after receiving updates from the server. Comparing with simpler methods, the only additional computation overhead, ....
> >
> > having to grid search over a hyperparameter space represents a big problem for a practical FL algorithm in my opinion.
> >
> > > Although paper (Kim, et al. 2023) proposes to define Normal-inverse Wishard (NIW) distribution and mixture models as hyper-priors over personalized posterior parameters, in the implementation, they simplify the NIW to "spiky Gaussian" by fixing the covariance to a singular identity matrix, as in their Equation (16). This means they fail to aggregate the covariances of the personalized posteriors ...
> >
> > Sure, they choose a variational family that doesn't capture covariance, but this seems to be the only methodological difference. The comment on the comparison between your work and the work of Hassan et al. 2023 does not make sense. You both have the goal of posterior inference of latent variables (or "distribution parameters" - I don't know what this means). the difference is that you have different algorithms and corresponding objectives to fit such a model.
> >
> > Again, thanks for your rebuttal. I will keep my score.

---

> > > ### Author Response · Authors · 2025-04-07
> > >
> > > We thank the reviewer for your further comments on our work. We would like to make a further clarification on the two points.
> > >
> > > >"having to grid search over a hyperparameter space represents a big problem for a practical FL algorithm":
> > >
> > > Grid search for our model is not costly, since the search space is only 2-dimensional with $\lambda_1$ and $\lambda_2$. Our comprehensive experiments shows that these regularization coefficients are critical to the superior performance for Bayesian personalized federated learning (pFL) in general. Turning them off hurts all the Bayesian pFL significantly. **Our work uncovered this major performance bottleneck of the previous methods. The small additional cost can thus be well justified.**
> > >
> > > >"Sure, they choose a variational family that doesn't capture covariance, but this seems to be the only methodological difference. The comment on the comparison between your work and the work of Hassan et al. 2023 does not make sense":
> > >
> > > - Please note that (Kim, et al. 2023)'s failing to aggregate the covariances of the personalized posteriors by fixing the covariance to a singular identity matrix actually leads to two fundamental difference in our methodologies. First, without modeling covariances, **their local models degenerate to non-Bayesian neural networks (i.e., regular feedforward neural networks with MAP point estimation)**, as shown in Equations (16) and (21). In contrast, our local models are fully Bayesian neural networks. Second, because they fix the hyperpriors, they failed to discover the relationship and limitations of the Bayesian pFL methods.
> > >
> > > - The latent variables introduced by (Hassan et al. 2023), as described in Section 4.1 Hierarchical Bayesian Neural Network for Heterogeneous Data, model only the weights of the first hidden layer of the local neural networks as random variables. They only provided MAP point estimates for the rest of the weights in the neural network architectures.
> > >
> > > We will clarify these difference in our camera-ready version. We would appreciate if Reviewer could reconsider your score on our work.

---

### Official Review · Reviewer_5j3i · 2025-03-12

**Overall Recommendation:** 3

**Summary:**

This paper proposes a novel hierarchical Bayesian inference framework for Personalized Federated Learning (PFL) by specifying a conjugate hyper-prior over the parameters of personalized posteriors. This approach enables the joint computation of a global posterior distribution for aggregation, alongside personalized posteriors at the local level. The proposed framework strikes an elegant balance between local personalization and global model robustness, offering a more efficient and scalable solution for PFL.

**Claims And Evidence:**

This paper proposes a novel hierarchical Bayesian inference framework for Personalized Federated learning. I believe that the claims made in the submission are well supported by evidence, but need further refinement (see comments)

Comments

1.Because the authors want to improve local personalized learning while ensuring the ability to generalize the global model, the baseline portion of the experiment should be compared more with the latest approaches to personalized federated learning.

2.The experiments in this paper are primarily conducted on the FMNIST, CIFAR-10, and CIFAR-100 datasets. It is recommended to evaluate the model’s performance on datasets with a larger number of categories, such as Tiny-ImageNet, to assess its generalizability. Additionally, in Table 1, the global model performs poorly on FMNIST when the number of clients is 100 and 200. It would be necessary for the authors to provide an explanation for this phenomenon.

3.The paper frequently uses the abbreviation 'BNN' without specifying its full form. For clarity and consistency, it would be helpful to define 'BNN' explicitly when it is first introduced.

4. A minor typo issue in the caption of Figure 1. "Our PFL framework is on the left, not the right."

**Essential References Not Discussed:**

The paper comprehensively discusses the prior work related to its main contributions, appropriately cites and compares relevant approaches in the area of bayesian federated learning and personalized federated learning, and does not omit any important references that will have a significant impact on the understanding of the proposed approach.

**Experimental Designs Or Analyses:**

1.Because the authors want to improve local personalized learning while ensuring the ability to generalize the global model, the baseline portion of the experiment should be compared more with the latest approaches to personalized federated learning.

2.The experiments in this paper are primarily conducted on the FMNIST, CIFAR-10, and CIFAR-100 datasets. It is recommended to evaluate the model’s performance on datasets with a larger number of categories, such as Tiny-ImageNet, to assess its generalizability. Additionally, in Table 1, the global model performs poorly on FMNIST when the number of clients is 100 and 200. It would be necessary for the authors to provide an explanation for this phenomenon.

**Methods And Evaluation Criteria:**

The method proposed in this paper preserves the client's personalized features and ensures the global generalization capability, but it needs further improvement.

**Other Comments Or Suggestions:**

I noticed a minor typo issue in the caption of Figure 1. Our PFL framework is on the left, not the right.

**Other Strengths And Weaknesses:**

Strengths:

1. By defining the conjugate super-prior on the distribution of personalized a posteriori parameters, the article achieves global model optimization while preserving the personalized features, and verifies the effectiveness of the method on three datasets.

2. The article demonstrates that the existing Bayesian PFL methods are special cases of its framework, and the optimization objectives of these methods can be obtained by introducing additional constraints, which provides a unified perspective for subsequent research.

Weaknesses:

1.In Equation (5), for the sake of computational simplification, does the method assume independence between global and local parameters? In practical scenarios, there is likely some degree of dependency between them.

2.Hierarchical Bayesian inference requires computing the global posterior distribution on the server and frequently synchronizing client parameters. When the number of clients K is large, this may introduce significant communication overhead.

**Questions For Authors:**

I'll reorganize all my queries here
1.Because the authors want to improve local personalized learning while ensuring the ability to generalize the global model, the baseline portion of the experiment should be compared more with the latest approaches to personalized federated learning.

2.The experiments in this paper are primarily conducted on the FMNIST, CIFAR-10, and CIFAR-100 datasets. It is recommended to evaluate the model’s performance on datasets with a larger number of categories, such as Tiny-ImageNet, to assess its generalizability. Additionally, in Table 1, the global model performs poorly on FMNIST when the number of clients is 100 and 200. It would be necessary for the authors to provide an explanation for this phenomenon.

3.The paper frequently uses the abbreviation 'BNN' without specifying its full form. For clarity and consistency, it would be helpful to define 'BNN' explicitly when it is first introduced.

4.In Equation (5), for the sake of computational simplification, does the method assume independence between global and local parameters? In practical scenarios, there is likely some degree of dependency between them. It would be beneficial to clarify whether this assumption affects the model's expressiveness.

5.Hierarchical Bayesian inference requires computing the global posterior distribution on the server and frequently synchronizing client parameters. When the number of clients K is large, this may introduce significant communication overhead.

**Relation To Broader Scientific Literature:**

This paper proposes a personalized federated learning method based on hierarchical Bayesian inference, which is of significance in helping the model adaptively adjust the trade-off between sharing and personalization among different clients by constructing a shared global prior and a personalized posterior.

**Theoretical Claims:**

No separate theoretical analysis section in the text.

---

> ### Author Rebuttal · Authors · 2025-03-31
>
> We thank Reviewer 5j3i for taking the time to provide detailed and constructive review on our paper.
>
> **"the baseline portion of the experiment should be compared more with the latest approaches to personalized federated learning"**: To the best of our knowledge, we included the latest competing Bayesian and non-Bayesian FL methods as the baselines. In addition, we show the performance comparison below with two pFL published in 2024. Please note that they are less comparable because both of them require additional datasets to pre-train the GMs.
> |Method|Number of clients|Cifar10 PM|Cifar100 PM
> |--|--|--|--|
> |FedBNN [1] |20|**79.9 &plusmn; 0.3**|57.3 &plusmn; 0.8
> |FedPPD [2] |10|61.86|53
> |Ours|50|77.1 &plusmn; 0.1| **66.6 &plusmn; 0.4**
>
> By aggregating information from even more clients (more clients, more challenging the FL task is) we still achieve comparable or better performance in Cifar100.
>
> **"evaluate the model’s performance on datasets with a larger number of categories, such as Tiny-ImageNet,.."**: As suggested by the reviewer, we conduct experiments on the Tiny-ImageNet dataset with 50 clients, where each client has unique local data consisting of 50 labels out of 200, and report the results below:
>
> |Methods| PM | GM
> |--|--|--|
> |FedAvg | -| 10.1 &plusmn; 0.7
> |pFedVEM|33.45 &plusmn; 0.2|18.5 &plusmn; 0.4
> |Ours| **42.4 &plusmn; 0.1**| **25.7 &plusmn; 0.1**
>
> It shows both PM and GM of our framework achieve the best performance. We will add the results in the camera-ready version.
>
> **"in Table 1, the global model performs poorly on FMNIST when the number of clients is 100 and 200.."** Our global model is the second best on FMNIST. This is mainly because the FMNIST dataset is relatively simple and does not have many non-IID features. Therefore, the advantages of personalization algorithms are not obvious. This can be verified by the fact that all the baselines including FedAvg achieves similar performance. We will clarify this in the camera-ready version.
>
> **Abbrevations and typos:** We will provide explanations for the abbrevations, and correct all the typos in the camera-ready version.
>
> **"does the method assume independence between global and local parameters?"**: As discussed  in Section 3.2 Line 126-128, the local parameters of client k’s personalized posterior $\theta_k$ are direct instantiations of the hyper-prior $p(\theta_g|\alpha)$. So they are dependent on the global parameters.
>
> **"Hierarchical Bayesian inference requires computing the global posterior distribution on the server and frequently synchronizing client parameters"**: As discussed in Section 3.3, we split the overall ELBO into server part and client part to avoid the potential communication overhead caused by frequently synchronizing client parameters. Because the client part of the ELBO can be optimized locally in multiple rounds after receiving updates from the server.
>
> We will clarify the points in the camera-ready version.
>
> **References**
>
> [1] Makhija, Disha, Joydeep Ghosh, and Nhat Ho. "A Bayesian Approach for Personalized Federated Learning in Heterogeneous Settings." Advances in Neural Information Processing Systems 37 (2024): 102428-102455.
>
> [2] Bhatt, Shrey, Aishwarya Gupta, and Piyush Rai. "Federated learning with uncertainty via distilled predictive distributions." Asian Conference on Machine Learning. PMLR, 2024.

---

### Official Review · Reviewer_DMYx · 2025-03-16

**Overall Recommendation:** 1

**Summary:**

The paper proposes a hierarchical Bayesian inference framework for personalized federated learning (PFL), addressing the issue of statistical heterogeneity across decentralized client datasets. The main conceptual contribution is introducing a conjugate hyperprior over personalized posterior parameters, allowing simultaneous inference of local personalized models and global aggregated models. This approach generalizes prior Bayesian methods for federated learning, aiming to balance local model personalization with global model robustness. Experimental results show competitive performance, especially in global model accuracy.

**Claims And Evidence:**

The paper's primary claim—that the hierarchical Bayesian approach effectively harnesses statistical heterogeneity—is supported through extensive empirical evaluation. However, the claim of significantly improved personalized model performance compared to existing methods is not convincingly demonstrated. While the global model does perform robustly, personalized models show only modest or comparable improvements.

**Essential References Not Discussed:**

To the best of my knowledge, the related work is well covered.

**Experimental Designs Or Analyses:**

Considering the existing body of work on Bayesian inference for federated learning, I find the novelty of the proposed approach limited, and the empirical results presented are not particularly strong. Specifically, the method does not significantly improve personalized model accuracy—the most crucial metric in personalized federated learning—compared to existing approaches. Performance improvements, if any, are minimal. Furthermore, the selected baselines do not explicitly require personalization, and better results could be achieved by training a single model with access to all data, suggesting that the primary challenge with the chosen datasets may lie in optimization rather than modeling methodology. Additionally, Bayesian methods generally demand significantly greater computational resources and present scalability issues. Consequently, non-Bayesian approaches, which can leverage much larger network sizes due to their efficiency, should also be included in the evaluations to provide a fairer comparison.

**Methods And Evaluation Criteria:**

The experimental design is generally sound. Authors evaluated the method across synthetic and real datasets (Fashion-MNIST, CIFAR-10, CIFAR-100), assessing statistical heterogeneity and scalability with varied client numbers. However, a significant limitation is that Bayesian methods inherently incur high computational costs, which was not explicitly compared against simpler, non-Bayesian baselines using larger networks or alternative optimization methods.

**Other Comments Or Suggestions:**

See my review above.

**Other Strengths And Weaknesses:**

See my review above.

**Questions For Authors:**

See my review above.

**Relation To Broader Scientific Literature:**

The main conceptual contribution is introducing a conjugate hyperprior over personalized posterior parameters, allowing simultaneous inference of local personalized models and global aggregated models. This approach generalizes prior Bayesian methods for federated learning, aiming to balance local model personalization with global model robustness. Experimental results show competitive performance, especially in global model accuracy.

**Theoretical Claims:**

I checked the theoretical derivations of the hierarchical Bayesian inference framework and the special-case reductions to existing methods (pFedBayes and pFedVEM). The derivations appear correct, clearly showing how previous methods can be recovered as special instances of the proposed model under specific constraints. No major theoretical issues were observed.

---

> ### Author Rebuttal · Authors · 2025-03-31
>
> We thank Reviewer DMYx for taking the time to provide a review on our paper.
>
> **"...personalized models show only modest or comparable improvements":** The results in Table 1 (Page 6) shows the personalized models (PMs) of our framework achieves significant better performance than the baseline methods across all the benchmark datasets. Our PM's performance is comparable with the baselines only on the synthetic data. This is because the synthetic data are relatively simple in terms of lower feature dimensionality and smaller dataset sizes than the benchmark data. All the baselines can perform well in terms of capturing the personalized information.
>
> **"Bayesian methods inherently incur high computational costs...:** This general claim about Bayesian inference is not necessarily applicable for FL problems. Figure 3 (Page 7) shows that our method achieves better performance with significant fewer communication rounds than other Bayes and non-Bayes FL approaches. The average local training time (seconds) per communication round in Cifar10 with all 100 clients available is
> |Methods|Time
> |--|--
> |non-Bayes (FedAvg)|21.42
> |non-Bayes Base, Bayes Head (pFedVEM)|43.14
> |full-Bayes (Ours,pFedBayes)|90.38
>
> So the overall computational cost is not necessarily higher than non-Bayes FL methods.
>
> **"I find the novelty of the proposed approach limited...":** The novel contributions of our work are
> - we propose a novel hierarchical Bayesian inference framework for personalized federated learning (pFL), introducing a hyper-prior over personalized posterior parameters;
> - we develop a stochastic variational inference scheme for the framework, enabling joint computation of global and local posteriors to balance local personalization and global robustness;
> - we subsume the existing Bayesian pFL methods as special cases of our general framework, shedding light on their relationships and limitations;
> - our framework significantly outperforms existing pFL methods in terms of both GMs and PMs across the benchmark datasets.
>
> We will clarify the novel contributions in the camera-ready version.
>
>
> **"the selected baselines do not explicitly require personalization, and better results could be achieved by training a single model with access to all data, suggesting that the primary challenge with the chosen datasets may lie in optimization rather than modeling methodology":**
> - The benchmark datasets we used are widely adopted for evaluating FL methods. Furthermore, as in Section 6.2, we applied common data partitioning approaches to examine various categories of statistical heterogeneity that exist in real-world FL scenarios including label distribution skew (i.e., clients possess varying label distributions), label concept drift (i.e., feature distributions differ among clients), and data quantity disparity (clients have different amounts of data). This is obviously a more challenging problem scenario than "training a single model with access to all data".
> - Figure 4 (Page 8) shows that our method provides greater benefits for the clients to participate in the collaborative training process in comparison with training a single model on its own or the competing FL approaches.
> - Although our work focuses on personalized FL, we included both personalized and non-personalized state-of-the-art FL methods as baselines for performance comparison.
>
> **"Bayesian methods generally demand significantly greater computational resources and present scalability issues...":** To the best of our knowledge, there is no evidence suggesting Bayesian FL suffers from "greater computational resources and present scalability issues". We validated and tested our framework with the largest number of clients (i.e., 200) among the FL literature reported on the image datasets. To further show the scalability of our method, we conduct experiments with a total of 1000 clients using the Cifar10 dataset, where each client has unique local data of 5 labels out of 10., and report the results below:
>
> |Methods| PM | GM
> |--|--|--|
> |FedAvg | -| 48.7 &plusmn; 1.1
> |pFedVEM| 59.0 &plusmn; 0.4|48.6 &plusmn; 0.9
> |Ours| **61.0 &plusmn; 0.3**| **52.3 &plusmn; 0.4**
>
> As shown in the table our method scales well with even 1000 clients and still outperfroms both Bayes FL (pFedVEM) and non-Bayes FL (FedAvg).
>
> **"non-Bayesian approaches, which can leverage much larger network sizes due to their efficiency, should also be included in the evaluations to provide a fairer comparison."** In our performance comparison in Table 1 and communication efficiency comparison in Figure 3, we reported state-of-the-art performance of both Bayes and non-Bayes FL methods. The non-Bayes FLs include FedAvg, pFedMe, FedRep, etc. The results shows that our method achieves significant better performance with less communication rounds. Using larger network sizes for non-Bayes FL will also incurs higher computational cost and not necessarily achieves better performance for non-IID FL data settings.

---

> > ### Comment · Reviewer_DMYx · 2025-04-09
> >
> > Thank you for the rebuttal. I appreciate the proposed approach and find it interesting. However, I am still not convinced by the provided evaluation. As mentioned in my review, the evaluation only considers a very limited set of networks (not even ResNet18) and a small number of communication rounds (≤ 100). This setup may be unfair to non-Bayesian methods, which typically perform fewer local computations and require more communication rounds to reach comparable or better accuracy. These additional rounds could come at a similar or even lower cost than the proposed Bayesian approach—as demonstrated in your rebuttal, FedAvg is more than 4x cheaper.
> >
> > Furthermore, I stand by my original point: **“the selected baselines do not explicitly require personalization, and better results could be achieved by training a single model with access to all data, suggesting that the primary challenge with the chosen datasets may lie in optimization rather than modeling methodology.”** For instance, FedAvg optimizes the same objective as a centralized model with access to all data. It is worth noting that we can simulate global SGD with IID client sampling by performing only one local step. The issue is that this would require a prohibitively large number of communication rounds. FedAvg mitigates this by using multiple local steps, which introduces client drift—making this fundamentally an optimization problem under a constrained communication budget. Thus, a model trained on all the data without federation remains a valid baseline and can be viewed as an upper bound on FedAvg performance.
> >
> > If this centralized baseline outperforms any global or personalized model, then personalization is not necessary because the modeling itself does not require it. Therefore, for a meaningful comparison, it is essential that the considered baselines genuinely require personalization.
> >
> > In addition, I find the method’s dependence on many hyperparameters—often inherited from multiple components—problematic. These hyperparameters are not necessarily consistent across experiments.
> >
> > In summary, while I like the direction and potential of the proposed method, the mentioned weaknesses—particularly the limited evaluation and questionable choice of baselines—prevent me from recommending a higher score. Thus, I maintain my current evaluation.

---

### Official Review · Reviewer_7w15 · 2025-03-17

**Overall Recommendation:** 4

**Summary:**

The paper introduces a hierarchical Bayesian inference framework for personalized federated learning (pFL) that specifies personalized posterior parameters, enabling joint computation of global and local posteriors to balance personalization and global robustness. While theoretically subsuming existing Bayesian pFL methods (e.g., `pFedBayes`, `pFedVEM`), it can produce superior global model (GM) and personalized model (PM) performances under statistical heterogeneity.

## Update after rebuttal
---
I updated the score to **accept** because the authors made a great effort to address all of my concerns.
I have summarized below how my evaluation changes after the rebuttal:
* The authors have faithfully performed additional experiments and theoretical analyses according to the review.
* Thus, I can happily conclude that both empirical and theoretical improvements are made during the rebuttal period.

**Claims And Evidence:**

### Main Claim
---
The proposed hierarchical Bayesian framework i) outperforms existing pFL methods in terms of both GM and PM performances by ii) effectively sharing a model weights while retaining local-specific knowledge, and iii) also subsumes existing Bayesian pFL methods.

### Breakdown
---
i) pFL Performance
* This main claim is well supported by empirical performances in Table 1 (Page 6) and Figure 2 (Page 5) even under practical setup, involving heterogeneity adjusted by $(\alpha, \beta)$ and partial client participation.
* In Figure 4, it shows better PM accuracy across varying local data sizes on benchmark datasets, indicating better collaboration benefits of the proposed method.

ii) Effective Sharing Strategy
* The authors adopted base-head separation, which is widely used in practice, for the effectiveness of pFL.
* However, only a simple separation has been provided, and none of the ablation (e.g. the order of updating, only sharing either the head or the base) or in-depth analysis/justification is provided.

iii) Subsumption
* In Section 4, the authors neatly expanded that their proposed method generalizes existing Bayesian pFL methods.
* Describing how such a subsumption helps or improves existing methods is more helpful in highlighting the proposed method, but such expositions are lacking in the current form.

**Essential References Not Discussed:**

N/A

**Experimental Designs Or Analyses:**

Although all experiments are limited to image classification benchmark datasets with a moderate number of clients (i.e. <200 clients), this is a common setup in the FL literature and is therefore acceptable.

**Methods And Evaluation Criteria:**

Yes, the proposed method and the evaluation criteria follow a typical scheme of pFL methods, and the authors properly simulated statistical heterogeneity due to label skewness and concept drift, both of which well reflect the real-world situation.

**Other Comments Or Suggestions:**

Please see 'Questions For Authors' section.

**Other Strengths And Weaknesses:**

Strengths
* The proposed method subsumes existing Bayesian pFL methods.
* The variational inference-based optimization is easy to be implemented in practice thanks to Algorithm 1.

Weaknesses
* There is lack of justification of why the hyper prior distribution should be half-normal distribution.
* The approximation of $q(\boldsymbol{\theta}_g;\phi)$ simplified Eq. 8, however, it can possibly affect to the overestimate of GM weights, high sensitivity to initialization, etc. Thus, proper correction method could be exploited, e.g., Laplace approximation, temperature scaling.
* The proposed method requires to tune the hyperparameter of hyper prior distribution, i.e., $\lambda_1, \lambda_2$.
* The proposed method has only been validated with a moderate number of clients, which may not fully reflect a large-scale cross-device FL setup.
* The proposed method only shows the pFL performance, and bypassed the strength of Bayesian inference (e.g., uncertainty quantification).

**Questions For Authors:**

* In what practical situation can we consider using the proposed method? The Bayesian perspective of pFL is intriguing in itself, and I believe that a complementary practical impact would also increase the validity of the proposed method. For example, authors could consider adding uncertainty quantification-related benchmark experiments under noisy-label scenario (e.g., see experimental setup of 'robustness to label noise' section in (Hahn, Jeong and Lee, 2022))
* Could the authors please provide any reasons of why the PM performance does not outperform other methods like PM did in Figure 2?
* How much local computation overhead is required for local BNN training, i.e., the computation of Eq. (10), compared to the non-Bayesian setup, e.g., `FedAvg`?
* Please also consider adding a comparison of the communication cost/complexity of the proposed method to that of `pFedBayes` and `pFedVEM`.
* I presume that the order in which the base and head are updated is important for pFL performance. In Algorithm 1, lines 807-823, the authors suggest updating the head first and the base later. Could you please provide simple comparative results when i) the base is updated first and the head later, when ii) only the base is updated and communicated, following (Collins et al., 2021) and (Oh et al., 2022)? If approach ii) shows decent performance, I think it can contribute to a lightweight Bayesian posterior update of the current proposed method.
* Is there any justification for choosing the half-normal distribution as the prior rather than the (inverse) Wishart distribution? Any supporting evidence (e.g. appropriateness for hierarchical Bayesian inference, empirical comparisons, etc.) would be appreciated.
* In the caption of Figure 1, 'Overview of our PFL framework based on hierarchical Bayesian inference (right)' -> 'Overview of our PFL framework based on hierarchical Bayesian inference (**left**)'
* Please avoid using $(\alpha,\beta)$ in adjusting the statistical heterogeneity as it overlaps with the notation of hyperparameter of a hyper prior distribution defined in Eq. 3.


> (Hahn, Jeong and Lee, 2022) Connecting Low-Loss Subspace for Personalized Federated Learning
> (Collins et al., 2021) Exploiting shared representations for personalized federated learning
> (Oh et al., 2022) FedBABU: Towards Enhanced Representation for Federated Image Classification

**Relation To Broader Scientific Literature:**

FL typically assumes a point estimate of a global model parameter. With the contribution of the paper, FL can be extended to find the posterior distribution of a global model parameter, which can contribute to Bayesian inference-related tasks.

**Theoretical Claims:**

Theoretical convergence analysis is absent, and only relies on empirical convergence results (Figure 3, 4 and 6).

---

> ### Author Rebuttal · Authors · 2025-03-30
>
> We thank Reviewer 7W15 for taking the time to provide a constructive and detailed review.
>
> **"..only a simple separation has been provided.."**: For fair comparison, the base-head setting in our framework follows (Collins et al., 2021) and pFedVEM (Zhu et al., 2023). As Reviewer suggested, we investigate the impact of the updating orders of the PM's base and head. The performance on CIFAR10 with 50, 100, and 200 clients are as below:
> |Updating Order|Sharing Component|50 Clients (PM)|50 Clients (GM)|100 Clients (PM)|100 Clients (GM)|200 Clients (PM)|200 Clients (GM)
> |--|--|--|--|--|--|--|--
> |Base Only (Oh et al., 2022)|Base Only|76.8±0.2|-|72.9±0.2|-| 68.3±0.4|-
> |Base->Head|Both Head & Base|76.7±0.4|65.1±0.5|72.6±0.8|60.6±0.6|68.3±0.4|57.3±0.6
> |Head->Base|Both Head & Base |**77.1±0.1**|**65.9±0.9**|**74.7±0.3**|**63.8±0.3**|**70.4±0.2**|**59.5±0.7**
>
> The results show that the setting in our paper, as in the third row, is the best one. Note that if we only share base or head, the global model will be incomplete.
>
> **"..how such a subsumption helps or improves existing methods.."**: The benefits from our general framework are as follows:
> - we proved that one of the constraints of the existing methods is that they implicitly set the regularization coefficients $\lambda_1$ and $\lambda_2$ to $0$. As in Table 2, our ablation study shows that $\lambda_1=\lambda_2=0$ leads to inferior performance.
> - Through the principled general framework we proposed, it shed light on the relationships and limitations of the existing methods.
> - we derived the relationship between the coefficients and the hyper-parameters, showing how they govern the statistical strength of the sharing process.
>
> **"..the strength of Bayesian inference..uncertainty quantification"**: Following (Hahn, Jeong and Lee, 2022) setting, we quantify prediction uncertainty under symmetric label flipping with a noise ratio of 0.2 on cifar10:
> |Methods|50 Clients: ECE (Acc)|100 Clients: ECE (Acc)
> |--|--|--
> |FedAvg (GM)|0.37±0.06 (36.8±0.9)|0.35±0.08 (37.9±2.5)
> |pFedVEM (PM)|**0.13±0.01** (70.1±0.4)|0.17±0.0 (68.0±0.4)
> |Ours (PM)| 0.15±0.0 (**72.6±0.4**)|**0.16±0.01 (70.3±0.3)**
>
> Our method's uncertainty quantification is comparable with pFedVEM but achieves higher accuracy.
>
> **"Theoretical convergence analysis is absent.."**: Evaluating the statistical convergence of the local personalized posteriors, we can show that the posterior mean $E[\theta_k|D]$ is in between the MLE estimate $\hat{\theta}_k $ and the global mean $\mu_g$: $E[\theta_k|D_k,\mu_g,\tau^2_g]=\alpha_k\mu_g+(1-\alpha_k)\hat{\theta}_k$ where the convergence speed towards the global mean is given by $\alpha_k=\frac{\sigma^2_k}{\sigma^2_k+\tau^2_g}$. It shows there is more shrinkage for clients with smaller measurement precision.
>
> **"..justification of..half-normal distribution"**: half-normal is commonly used as a prior over the standard deviation of Gaussian. As a special case of inverse-Wishard dist., conjugate over Gaussian variance, half-normal is computationally more stable and more efficient.
>
> **"The approximation of $q(\theta_g|\phi)$.."**: Simplifying $q(\theta_g|\phi)$ to a delta function improves communication efficiency without sacrificing performance. Using the delta function, we only need to broadcast a single $\theta_g$ value back to the clients. With the original variational distribution, it needs to broadcast a set of samples to each client.
>
> **"..requires to tune the hyperparameters..**: Although tuning $\lambda_1$ and $\lambda_2$ incurs additional computation overhead as discussed in Appendix A.4, the performance gain is statistically significant across benchmarks.
>
> **"..validation with a moderate number of clients.."**: We validated our framework with the largest number of clients (i.e., 200) among the baseline methods on the image datasets (e.g., pFedBayes with maximum 20 clients, FedAvg 100 clients, and pFedVEM 200 clients). As Reviewer suggested, we conduct experiments with 1000 clients on Cifar10 as below:
> |Methods| PM | GM
> |--|--|--|
> |FedAvg|-|48.7±1.1
> |pFedVEM|59.0 ± 0.4|48.6±0.9
> |Ours|**61.0 ± 0.3**| **52.3±0.4**
>
> as shown above, our method scales well with large client size.
>
> **"why the PM performance does not outperform other methods..in Figure 2"**: The synthetic datasets are relatively simple in terms of lower feature dimensionality and smaller dataset sizes. All the baselines can perform well in terms of capturing the non-IID information.
>
> **"local computation overhead..for local BNN training"**: Average local training times (seconds) of BNNs per communication round on Cifar10 with all 100 clients available are:
> |Methods|Time
> |--|--
> |non-Bayes (FedAvg)|21.42
> |non-Bayes Base, Bayes Head (pFedVEM)|43.14
> |full-Bayes (Ours, pFedBayes)|90.38
>
> **"comparison of the communication cost..":** the communication cost comparison was reported in Figure 3 and in Section 6.2.1.
>
> We will clarify the above points and correct typos and notations in the camera-ready version.

---

> > ### Comment · Reviewer_7w15 · 2025-04-03
> >
> > I appreciate the detailed responses from the authors.
> >
> > ###  Why Not Lower Score
> > ---
> > As most of my concerns are addressed, I have **raised the score to weak accept.**
> > Based on the rebuttals, please consider adding or supplementing the following in the revised manuscript:
> > - More emphasis on subsumption as one of the main contributions
> > - Empirical scalability even under large number of clients (i.e., $K=1,000$)
> > - Discussion of the limitation/weakness of the proposed method in the practical scenario, i.e., increased local computational overhead, compared to non-Bayesian FL methods
> > - Appeal to the strength of the Bayesian approach more with empirical results for the practical FL scenario where label noise exists
> > - Supportive empirical evidence for the chosen update order (in the appendix)
> > I sincerely hope that these will help improve the presentation and communication of the research results.
> >
> > ###  Why Not Higher Score
> > ---
> > Please also note that there's room for additional score raising as long as **one remaining concern** is answered.
> > - Pertaining to the statistical convergence that the authors answered in the rebuttal, could the authors please provide more expositions on the statistical convergence analysis? (e.g., detailed procedure or even proof sketch)
> > - What does 'more shrinkage' mean in this context? Please clarify.

---

> > > ### Author Response · Authors · 2025-04-04
> > >
> > > We thank Reviewer 7w15 for your further comment on our work and raising the score. We will follow Reviewer's suggestion, adding and supplementing the details and additional experiments listed in the comment to our camera-ready version.
> > >
> > > We provide the analysis about the statistical convergence of our model below.
> > >
> > > After computing the approximate global posterior in Equation (4) via stochastic variational inference (i.e., Section 3.3), we analyze the convergence of the personalized distribution for each client $k$: $p(\mathcal{W}_k|\theta_k)$, where $\theta_k=\\{\mu_k,\sigma^2_k\\}$ are the parameters of the client $k$'s personalized posterior, as in Section 3.2, and $\mathcal{W}_k$ denotes the weights of the local model (i.e., Bayesian neural networks).
> > >
> > > Specifically, we analyze the convergence behavior of the personalized distributions towards the global mean $\bar{\mu}=E[\mu_g|\mathcal{W}]$, which is close to the pooled estimate $\bar{\mathcal{W}}$. Let the hyperparameters $\alpha$ to be fixed, and we adopt the approximation to the global posterior $p(\mu_g,\sigma^2_g|\mathcal{W})=\delta(\mu_g-\bar{\mu})\delta(\sigma^2_g-\bar{\sigma}^2)$ for simplicity, then the marginal distribution of the personalized distribution parameters is:
> > >
> > > \begin{align}
> > > p(\theta_k|\mathcal{W})&=\int p(\theta_k|\mathcal{W}_k,\theta_g)p(\theta_g|\mathcal{W})d\theta_g \\\\
> > > &\approx p(\theta_k|\mathcal{W}_k,\bar{\mu},\bar{\sigma}^2)
> > > \end{align}
> > > note that $\theta_g=\\{\mu_g,\sigma^2_g\\}$. Meanwhile, the above marginal posterior can be expressed as
> > > \begin{align}
> > > p(\theta_k|\mathcal{W}_k,\bar{\mu},\bar{\sigma}^2)&\propto p(\mathcal{W}_k|\theta_k)\cdot p(\theta_k|\bar{\mu},\bar{\sigma}^2)\\\\
> > > &=\mathcal{N}(\mathcal{W}_k|\theta_k)\cdot\mathcal{N}(\theta_k|\bar{\mu},\bar{\sigma}^2)\\\\
> > > &=\exp(-\frac{(\hat{\theta}_k-\theta_k)^2}{\sigma^2_k})\cdot\exp(-\frac{(\theta_k-\bar{\mu})^2}{\bar{\sigma}^2})\\\\
> > > &=\exp(-(\frac{1}{2\sigma^2_k}+\frac{1}{2\bar{\sigma}^2})\theta^2_k+(\frac{\hat{\theta}_k}{\sigma^2_k}+\frac{\bar{\mu}}{\bar{\sigma}^2})\theta_k-(\frac{\hat{\theta}^2_k}{2\sigma^2_k}+\frac{\bar{\mu}^2}{2\bar{\sigma}^2}))
> > > \end{align}
> > > where $\hat{\theta}_k=\bar{\mathcal{W}}_k$ denotes the local MLE estimate. Since the marginal posterior $p(\theta_k|\mathcal{W}_k,\bar{\mu},\bar{\sigma}^2)=\mathcal{N}(\theta_k|E[\theta_k|\mathcal{W}],Var[\theta_k|\mathcal{W}])$ is a Gaussian due to conjugacy, by matching the coefficients in the quadratic and linear terms of the two expressions, we have the variance of the marginal distribution as
> > > \begin{align}
> > > \frac{1}{2Var[\theta_k|\mathcal{W}]}&=\frac{1}{2\sigma^2_k}+\frac{1}{2\bar{\sigma}^2}\\\\
> > > Var[\theta_k|\mathcal{W}]&=\frac{\sigma^2_k\bar{\sigma}^2}{\sigma^2_k+\bar{\sigma}^2}
> > > \end{align}
> > > and the mean of the marginal distribution as
> > > \begin{align}
> > > \frac{E[\theta_k|\mathcal{W}]}{Var[\theta_k|\mathcal{W}]}&=\frac{\hat{\theta}_k}{\sigma^2_k}+\frac{\bar{\mu}}{\bar{\sigma}^2}\\\\
> > > E[\theta_k|\mathcal{W}]&=(\frac{\hat{\theta}_k}{\sigma_k}+\frac{\bar{\mu}^2}{\bar{\sigma}^2})\cdot Var[\theta_k|\mathcal{W}]\\\\
> > > &=\beta_k\bar{\mu}+(1-\beta_k)\hat{\theta}_k
> > > \end{align}
> > > where $\beta_k=\frac{\sigma^2_k}{\sigma^2_k+\bar{\sigma}^2}$. It indicates that the mean of the personalized distribution parameters: $E[\theta_k|\mathcal{W}]$ lies in between the local MLE estimate $\hat{\theta}_k=\bar{\mathcal{W}}_k$ and the global mean $\bar{\mu}$. The convergence of the personalized parameters towards the global mean is governed by $\beta_k$. Thus, we see that there is larger convergence for clients with smaller measurement precision (e.g., due to smaller data sizes or noisy labels), since $\beta_k\rightarrow 1$ as $\sigma^2_k\rightarrow\infty$.
> > >
> > > In this context, 'more shrinkage' refers to the convergence behaviors towards the global model.

---

### Decision · Program_Chairs · 2025-05-01

**Decision:**

Accept (poster)

**Comment:**

This paper introduces a Hierarchical Bayesian approach to personalized federated learning, with hyperpriors over the local Gaussian Bayesian neural networks.

Reviewers diverge in their view of this work. Reviewer 7w15 increased their score from 2 to 4 during the rebuttal, as they felt like the authors addressed their concerns such as computation cost and additional benefits of being Bayesian.

Reviewer DMYx recommends reject, but in my opinion, the authors have rebutted the reviewer's concerns on computation cost of the method, and the point about personalized vs global models is not so important. The authors can consider adding a global baseline (as if all data was available in one place), which would be an upper bound on global performance, and which I think all personalized FL papers should have anyway (to see what room there is for benefits of having personalized models). They (and other reviewers) point out the additional cost of tuning two new hyperparameters but I think that this is a limitation of the method that I commend the authors for explicitly mentioning and including an ablation.

Reviewers 5j3i and kx6d stuck with their scores after rebuttal but I think the authors have mostly rebutted their points. The paper will be stronger after including the additional results and discussion as promised by the authors, including comparison to 2 works as mentioned by reviewer kx6d, computation time results, and additional results with larger number of clients and larger benchmark.

Overall, I think the authors can more clearly give intuition as to why the existing methods that are part of their framework are worse than their method. They provide some justification in the rebuttal about lambda1=lambda2=0 but this was not sufficiently concise for me, and this can be improved for the camera-ready version.